# Maximum Likelihood Learning of Energy-Based Models for Simulation-Based Inference

## Abstract

We introduce two synthetic likelihood methods for Simulation-Based Inference (SBI), to conduct either amortized or targeted inference from experimental observations when a high-fidelity simulator is available. Both methods learn a conditional energy-based model (EBM) of the likelihood using synthetic data generated by the simulator, conditioned on parameters drawn from a proposal distribution. The learned likelihood can then be combined with any prior to obtain a posterior estimate, from which samples can be drawn using MCMC. Our methods uniquely combine a flexible Energy-Based Model and the minimization of a KL loss: this is in contrast to other synthetic likelihood methods, which either rely on normalizing flows, or minimize score-based objectives; choices that come with known pitfalls. Our first method, Amortized Unnormalized Neural Likelihood Estimation (AUNLE), introduces a tilting trick during training that allows to significantly lower the computational cost of inference by enabling the use of efficient MCMC techniques. Our second method, Sequential UNLE (SUNLE), utilizes a new conditional EBM learning technique in order to re-use simulation data and improve posterior accuracy on a specific dataset. We demonstrate the properties of both methods on a range of synthetic datasets, and apply them to a neuroscience model of the pyloric network in the crab, matching the performance of other synthetic likelihood methods at a fraction of the simulation budget.

## 1 Introduction

Simulation-based modeling expresses a system as a *probabilistic program* (Ghahramani, 2015), which describes, in a mechanistic manner, how samples from the system are drawn given the parameters of the said system. This probabilistic program can be concretely implemented in a computer - as a *simulator* - from which synthetic parameter-samples pairs can be drawn. This setting is common in many scientific and engineering disciplines such as stellar events in cosmology (Alsing et al., 2018; Schafer & Freeman, 2012), particle collisions in a particle accelerator for high energy physics (Eberl, 2003; Sjöstrand et al., 2008), and biological neural networks in neuroscience (Markram et al., 2015; Pospischil et al., 2008). Describing such systems using a probabilistic program often turns out to be easier than specifying the underlying probabilistic model with a tractable probability distribution. We consider the task of *inference* for such systems, which consists in computing the posterior distribution of the parameters given observed (non-synthetic) data. When a likelihood function of the simulator is available alongside with a prior belief on the parameters, inferring the posterior distribution of the parameters given data is possible using Bayes' rule. Traditional inference methods such as variational techniques (Wainwright & Jordan, 2008) or Markov Chain Monte Carlo (Andrieu et al., 2003) can then be used to produce approximate posterior samples of the parameters that are likely to have generated the observed data. Unfortunately, the likelihood function of a simulator is computationally intractable in general, thus making the direct application of traditional inference techniques unusable for simulation-based modelling.

*Simulation-Based Inference* (SBI) methods (Cranmer et al., 2020) are methods specifically designed to perform inference in the presence of a simulator with an intractable likelihood. These methods repeatedly generate synthetic data using the simulator to build an estimate of

the posterior, that either can be used for any observed data (resulting in a so-called *amortized* inference procedure) or that is *targeted* for a specific observation. While the accuracy of inference increases as more simulations are run, so does computational cost, especially when the simulator is expensive, which is common in many physics applications (Cranmer et al., 2020). In high-dimensional settings, early simulation-based inference techniques such as Approximate Bayesian Computation (ABC) (Marin et al., 2012) struggle to generate high quality posterior samples at a reasonable cost, since ABC repeatedly rejects simulations that fail to reproduce the observed data (Beaumont et al., 2002). More recently, model-based inference methods (Wood, 2010; Papamakarios et al., 2019; Hermans et al., 2020; Greenberg et al., 2019), which encode information about the simulator via a parametric density (-ratio) estimator of the data, have been shown to drastically reduce the number of simulations needed to reach a given inference precision (Lueckmann et al., 2021). The computational gains are particularly important when comparing ABC to *targeted* SBI methods, implemented in a *multi-round* procedure that refines the estimation of the model around the observed data by sequentially simulating data points that are closer to the observed ones (Greenberg et al., 2019; Papamakarios et al., 2019; Hermans et al., 2020).

Previous model-based SBI methods have used their parametric estimator to learn the likelihood (e.g. the conditional density specifying the probability of an observation being simulated given a specific parameter set, Wood 2010; Papamakarios et al. 2019; Pacchiardi & Dutta 2022), the likelihood-to-marginal ratio (Hermans et al., 2020), or the posterior function directly (Greenberg et al., 2019). We focus in this paper on likelihood-based (also called Synthetic Likelihood; SL, in short) methods, of which two main instances exist: (Sequential) Neural Likelihood (Papamakarios et al., 2019), which learns a likelihood estimate using a normalizing flow trained by optimizing a Maximum Likelihood (ML) loss; and Score Matched Neural Likelihood (Pacchiardi & Dutta, 2022), which learns an unnormalized (or *Energy-Based*, LeCun et al. 2006) likelihood model trained using conditional score matching. Recently, SNL was applied successfully to challenging neural data (Deistler et al., 2021). However, limitations still remain in the approaches taken by both SNL and SMNL. One the one hand, flow-based models may need to use very complex architectures to properly approximate distributions with rich structure such as multi-modality (Kong & Chaudhuri, 2020; Cornish et al., 2020). On the other hand, score matching, the objective of SMNLE, minimizes the Fisher Divergence between the data and the model, a divergence that fails to capture important features of probability distributions such as mode proportions (Wenliang & Kanagawa, 2020; Zhang et al., 2022). This is unlike Maximimum-Likelihood based-objectives, whose maximizers satisfy attractive theoretical properties (Bickel & Doksum, 2015).

**Contributions.** In this work, we introduce *Amortized Unnormalized Likelihood Neural Estimation* (AUNLE), and *Sequential UNLE*, a pair of SBI Synthetic Likelihood methods performing respectively sequential and targeted inference. Both methods learn a Conditional Energy Based Model of the simulator's likelihood using a Maximum Likelihood (ML) objective, and perform MCMC on the posterior estimate obtained after invoking Bayes' Rule. While posteriors arising from conditional EBMs exhibit a particular form of intractability called *double intractability*, which requires the use of tailored MCMC techniques for inference, we train AUNLE using a new approach which we call *tilting*. This approach automatically removes this intractability in the final posterior estimate, making AUNLE compatible with standard MCMC methods, and significantly reducing the computational burden of inference. Our second method, SUNLE, departs from AUNLE by using a new training technique for conditional EBMs which is suited when the proposal distribution is not analytically available. While SUNLE returns a doubly intractable posterior, we show that inference can be carried out accurately through robust implementations of doubly-intractable MCMC methods. We demonstrate the properties of AUNLE and SUNLE on an array of synthetic benchmark models (Lueckmann et al., 2021), and apply SUNLE to a neuroscience model of the crab *Cancer borealis*, increasing posterior accuracy over prior art while needing only a fraction of the simulations required by the most efficient prior method (Glöckler et al., 2021).

## 2 BACKGROUND

Simulation Based Inference (SBI) refers to the set of methods aimed at estimating the posterior $p(\theta|x_o)$ of some unobserved parameters $\theta$ given some observed variable $x_o$ recorded from a physical system, and a prior $p(\theta)$. In SBI, one assumes access to a simulator

|  SMNLE | NLE | **AUNLE (Ours)** | Ground Truth |

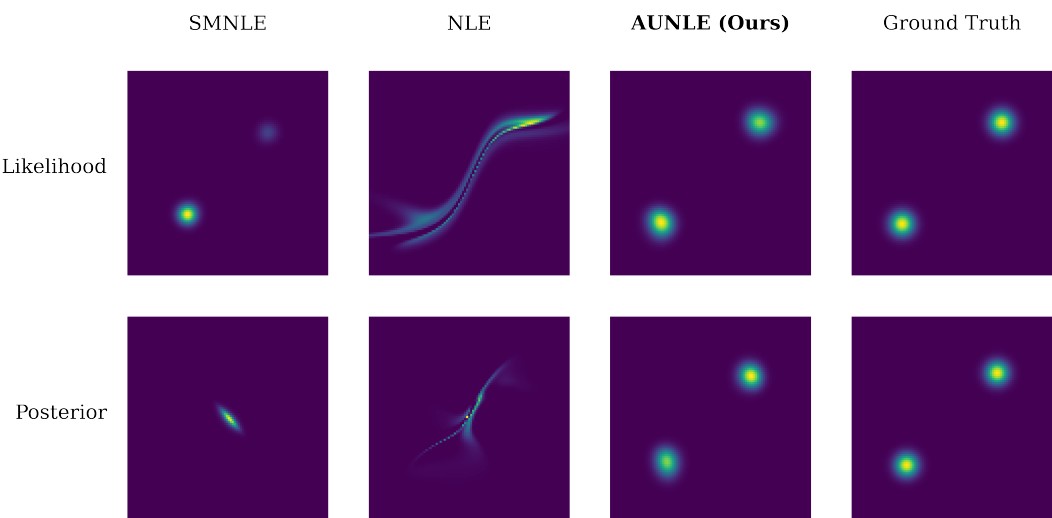

Figure 1: Performance of SMNLE, NLE and AUNLE traing using a simulator with a bimodal likelihood $p(x|\theta)$, and a gaussian prior $p(\theta)$ using 1000 samples. Top: Simulator likelihood $p(x|\theta_0)$ for some fixed $\theta_0$. Bottom: posterior estimate.

$G : (\theta, u) \longmapsto y = G(\theta, u)$, from which samples $y|\theta$ can be drawn, and whose associated likelihood $p(y|\theta)$ accurately matches the likelihood $p(x|\theta)$ of the physical system of interest. Here, $u$ represents draws of all random variables involved in performing draws of $x|\theta$. By a slight abuse of notation, we will not distinguish between the *physical* random variable $x$ representing data from the *physical* system of interest, and the *simulated* random variable $y$ draw from the simulator: we will use $x$ for both. The complexity of the simulator (Cranmer et al., 2020) prevents access to a simple form for the likelihood $p(x|\theta)$, making standard Bayesian inference impossible. Instead, SBI methods perform inference by drawing parameters from a proposal distribution $\pi(\theta)$, and use these parameters as inputs to the simulator $G$ to obtain a set of simulated pairs $(x, \theta)$ which they use to compute a posterior estimate of $p(\theta|x)$. Specific SBI submethods have been designed to handle separately the case of *amortized* inference, where the practitioner seeks to obtain a posterior estimate valid for any $x_o$ (which might not be known a priori), and *targeted* inference, where the posterior estimate should maximize accuracy for a specific observed variable $x_o$. While amortized inference methods set their proposal distribution $\pi$ to be the prior $p$, targeted inference methods iteratively refine their proposal $\pi$ to focus their simulated observations around the targeted $x_o$ through a *sequence* of simulation-training rounds (Papamakarios et al., 2019).

## 2.1 (CONDITIONAL) ENERGY-BASED MODELS.

Energy-Based Models (LeCun et al., 2006) are unnormalized probabilistic models of the form

$$q_\psi(x) = \frac{e^{-E_\psi(x)}}{Z(\psi)}, \quad Z(\psi) = \int e^{-E_\psi(x)} \mathrm{d}x,$$

where $Z(\psi)$ is the intractable normalizing constant of the model, and $E_\psi$ is called the *energy function*, usually set to be a neural network with weights $\psi$. By directly modelling the density $p(x)$ of the data through a flexible energy function, simple EBMs can capture rich geometries and multi-modality, whereas other model classes such a normalizing flows may require a more complex architecture (Cornish et al., 2020). The flexibility of EBMs comes at the cost of having an intractable density $q_\psi(x)$ due to the presence of the normalizer $Z(\psi)$, increasing the challenge of both training and sampling. In particular, an EBM's log-likelihood $\log q_\psi$ and associated gradient $\nabla_\psi \log q_\psi$ both contain terms involving the (intractable) normalizer $Z(\psi)$:

$$\log q_\psi(x) = -E_\psi(x) - \overbrace{\log Z(\psi)}^{\text{intractable}}, \quad \nabla_\psi \log q_\psi(x) = -\nabla_\psi E_\psi(x) + \overbrace{\mathbb{E}_{x \sim q_\psi} \nabla_\psi E_\psi(x)}^{\text{intractable}}. \quad (1)$$

making *exact* training of EBMs via Maximum Likelihood impossible. Approximate likelihood optimization can be performed using a Gradient-Based algorithm where at each iteration

$k$, the intractable expectation (under the EBM $q_{\psi_k}$) present in $\nabla_\psi \log q_{\psi_k}$ is replaced by one under a *particle approximation* $\widehat{q} = \frac{1}{N}\sum_{i=1}^{N} w_i \delta_{y_i}$ of $q_\psi$. The particles $y^{(i)}$ forming $\widehat{q}$ are traditionally set to be samples from a MCMC chain with invariant distribution $q_{\psi_k}$, with uniform weights $w_i = \frac{1}{N}$, while recent work on EBM for high-dimensional image data uses an adaptation of Langevin Dynamics (Raginsky et al., 2017; Du & Mordatch, 2019; Nijkamp et al., 2019; Kelly & Grathwohl, 2021). We outline the traditional ML learning procedure for EBM in Algorithm 2, where `make_particle_approx`$(q, \hat{q}_0)$ is a generic routine producing a particle approximation of a target unnormalized density $q$ and an initial particle approximation $\hat{q}_0$.

Energy-Based Models are naturally extended to both *joint* EBMs $q_\psi(\theta, x) = \frac{e^{-E_\psi(\theta, x)}}{Z(\psi)}$ (Kelly & Grathwohl, 2021; Grathwohl et al., 2020) and *conditional* EBMs (CEBMs Khemakhem et al. 2020; Pacchiardi & Dutta 2022) of the form:

$$q_\psi(x|\theta) = \frac{e^{-E_\psi(x,\theta)}}{Z(\theta,\psi)}, \quad Z(\theta;\psi) = \int e^{-E_\psi(x,\theta)}\mathrm{d}x. \tag{2}$$

Unlike joint and standard EBMs, conditional EBMs define a family of conditional densities $q_\psi(x|\theta)$, each of which is endowed with an intractable normalizer $Z(\theta,\psi)$.

## 2.2 Synthetic Likelihood Methods for SBI

Synthetic Likelihood (SL) methods (Wood, 2010; Papamakarios et al., 2019; Pacchiardi & Dutta, 2022) form a class of SBI methods that learn a conditional density model $q_\psi(x|\theta)$ of the unknown likelihood $p(x|\theta)$ for every possible pair of observations and parameters $(x, \theta)$. The set $\{q_\psi(x|\theta),\ \psi \in \Psi\}$ is a model class parameterised by some vector $\psi \in \Psi$, which recent methods set to be a neural network with weights $\psi$. We describe the existing Neural SL variants to date.

**Neural Likelihood Estimation** (NLE, Papamakarios et al. 2019) sets $q_\psi$ to a (normalized) flow-based model, and is optimized by maximizing the *average conditional log-likelihood* $\mathbb{E}_{\pi(\theta)p(x|\theta)}\log q_\psi(x|\theta)$. NLE performs inference by invoking Bayes' rule to obtain an unnormalized posterior estimate $p_\psi(\theta|x) = \frac{q_\psi(x|\theta)p(\theta)}{\int q_\psi(x|\theta)p(\theta)\mathrm{d}\theta} \propto p(\theta)q_\psi(x|\theta)$ from which samples can be drawn either using MCMC, or Variational Inference (Glöckler et al., 2021).

**Score Matched Neural Likelihood Estimation** (SMNLE, Pacchiardi & Dutta 2022) models the unknown likelihood using a conditional Energy-Based Model $q_\psi(x|\theta)$ of the form of Equation (2), trained using a score matching objective adapted for conditional density estimation. The use of an unnormalized likelihood model makes the posterior estimate obtained via Bayes' Rule known up to a $\theta$-dependent term:

$$q_\psi(\theta|x) \propto p(\theta)q_\psi(x|\theta) \propto \frac{e^{-E_\psi(x,\theta)}p(\theta)}{\underbrace{Z(\theta)}_{\text{intractable}}}, \quad Z(\theta) = \int e^{-E_\psi(x,\theta)}\mathrm{d}x. \tag{3}$$

Posteriors of this form are called *doubly intractable* posteriors (Møller et al., 2006). In the case where the likelihood $q_\psi(x|\theta)$ can be sampled from, Møller et al. (2006); Murray et al. (2006) have proposed tractable MCMC methods that draw an auxiliary variable $y \sim q_\psi(x|\theta)$ at every iteration to compute the acceptance probability of the proposed sample. Importantly, these MCMC methods still admit $q_\psi(\theta|x)$ as their invariant distribution, making inference as exact as in standard MCMC methods. In the case of SMNLE however, $q_\psi(x|\theta)$ cannot be tractably sampled from; SMNLE instead uses an *approximate doubly intractable* method, which replaces the exact sample $y$ by the result of an MCMC chain with invariant distribution $q_\psi(x|\theta)$. Even though this variant introduces an additional approximation not present in standard (*"singly" intractable*) MCMC algorithms, the distance between the true posterior and the distributions of the MCMC samples can be bounded under specific assumptions (Alquier et al., 2016).

Both the likelihood objective of NLE and the score-based objective of SMNLE do not involve the analytic expression of the proposal $\pi$, making it easy to adapt these methods for either

amortized or targeted inference. To address the limitations of both methods mentioned in the introduction, we next propose a method that combines the use of flexible Energy-Based Models as in SMNLE, while being optimized using a likelihood loss as in NLE.

## 3 Unnormalized Neural Likelihood Estimation

In this section, we present our two methods, Amortized-UNLE and Sequential-UNLE. Both AUNLE and SUNLE approximate the unknown likelihood $p(x|\theta)$ for any possible pair of $(x, \theta)$ using a *conditional* Energy-Based Model $q_\psi(x|\theta)$ as in Equation (2), where $E_\psi$ is some neural network. Additionally, AUNLE and SUNLE are both trained using a likelihood-based loss; however, the training objectives and inference phases differ to account for the specificities of amortized and targeted inference, as detailed below.

### 3.1 Amortized UNLE

Given a likelihood model $q_\psi(x|\theta)$, a natural learning procedure would involve fitting a model $q_\psi(x|\theta)\pi(\theta)$ of the true "joint synthetic" distribution $\pi(\theta)p(x|\theta)$, as NLE does. However, we show that using an alternative – tilted – version of this model allows to compute a posterior that is more tractable than the ones computed by other SL methods relying on conditional EBMs such as SMNLE (Pacchiardi & Dutta, 2022). Our method, AUNLE, fits a joint probabilistic model $q_{\psi,\pi}$ of the form:

$$q_{\psi,\pi}(x, \theta) := \frac{\pi(\theta)e^{-E_\psi(x,\theta)}}{Z_\pi(\psi)}, \quad Z_\pi(\psi) = \int \pi(\theta)e^{-E_\psi(x,\theta)}\mathrm{d}x\mathrm{d}\theta. \tag{4}$$

by maximizing its log-likelihood $\mathcal{L}_a(\psi) := \mathbb{E}_{\pi(\theta)p(x|\theta)}[\log q_{\psi,\pi}(x, \theta)]$ using an instance of Algorithm 2. The gain in tractability offered by AUNLE is a direct consequence of the following proposition, its joint model.

**Proposition 1.** *Let $\mathcal{P}_\psi := \{q_\psi(\cdot|\theta), \ \psi \in \Psi\}$, and $q_\psi \in P_\psi$. Then we have:*

- *(likelihood modelling) $q_{\psi,\pi}(x|\theta) = q_\psi(x|\theta)$*

- *(joint model tilting) $q_{\psi,\pi}(x, \theta) = f(\theta)\pi(\theta)q_\psi(x|\theta)$, for $f(\theta) := Z(\theta, \psi)/Z_\pi(\psi)$*

- *($(Z, \theta)$-uniformization) If $p(\cdot|\theta) \in \mathcal{P}_\psi$, then the $\psi^\star$ minimizing $\mathcal{L}_a(\psi)$ satisfies: $q_\psi(x|\theta) = p(x|\theta)$, and $Z(\theta, \psi^\star) = Z_\pi(\psi^\star)$.*

*Proof.* The first point follows by holding $\theta$ fixed in $q_{\psi,\pi}(x, \theta)$. To prove the second point, notice that $q_{\psi,\pi}(x, \theta) = \frac{Z(\theta,\psi)}{Z(\theta,\psi)}\frac{\pi(\theta)e^{-E(x,\theta)}}{Z_\pi(\psi)} = \frac{Z(\theta,\psi)}{Z_\pi(\psi)}\pi(\theta)\frac{e^{-E(x,\theta)}}{Z(\theta,\psi)}$. For the last point, note that at the optimum, we have that $q_{\psi^\star,\pi}(x, \theta) = \pi(\theta)p(x|\theta)$. Integrating out $x$ on both sides of the equality yields $f(\theta)\pi(\theta) = \pi(\theta)$, proving the result. $\square$

Proposition 1 shows that AUNLE indeed learns a likelihood model $q_\psi(x|\theta)$ through a joint model $q_{\psi,\pi}$ *tilting* the prior $\pi$ with $f(\theta)$. Importantly, this tilting guarantees that the optimal likelihood model will have a normalizing function $Z(\theta; \psi)$ constant (or *uniform*) in $\theta$, reducing AUNLE's posterior to a standard unnormalized posterior $q_{\psi^\star}(\theta|x) = p(\theta)\frac{e^{-E_{\psi^\star}(\theta,x)}}{Z_\pi(\psi^\star)}$, from which samples can be drawn using classical MCMC techniques, as for NLE. AUNLE's posterior contrasts with the posterior of SMNLE (Pacchiardi & Dutta, 2022), an amortized SBI method which also computes a posterior using a conditional EBM of the likelihood, but that remains *doubly intractable*, as discussed in Section 2. The gain in tractability of AUNLE's posterior is beneficial from an inference accuracy standpoint as it removes the need to use an otherwise *approximate* doubly-intractable technique when performing inference. Importantly, such a property is also beneficial from a computational cost standpoint, since approximate doubly-intractable methods require running an (*inner*) MCMC chain with target $q_{\psi^\star}(x|\theta)$ for every iteration of the (*outer*) MCMC chain with target $q_{\psi^\star}(\theta|x)$, roughly squaring the computational cost of standard MCMC methods. This computational advantage is all the more important since AUNLE returns an *amortized* posterior, valid for any observed data $x_o$, and which may be thus sampled from more than once. We confirm in Appendix B.3 that

the $(Z, \theta)$-uniformization of AUNLE's posterior, which is only guaranteed *in a well-specified setting, at the true optimum $\psi^\star$*, holds well in practice.

---

**Algorithm 1** Amortized-UNLE

**Input:** prior $p(\theta)$, simulator $G$, budget $N$
**Output:** Posterior estimate $q_\psi(\theta|x)$
**Initialize** $\psi_0, q_{\psi_0, \pi} \propto e^{-E_{\psi_0}(x,\theta)} \pi(\theta)$
$\qquad \pi = p$
**for** $i = 0, \ldots, N$ **do**
$\qquad$ Draw $\theta \sim \pi$, $x \sim G(\theta, \cdot)$
$\qquad$ Add $(\theta, x)$ to $\mathcal{D}$
**end for**
Get $\psi^\star := \texttt{maximize\_ebm\_log\_l}(\mathcal{D}, \psi_0)$
Set $q_{\psi^\star}(\theta|x) := e^{-E_{\psi^\star}(x,\theta)} p(\theta)$
Infer using MCMC on $q_{\psi^\star}(\theta|x)$

---

**Algorithm 2** $\texttt{maximize\_ebm\_log\_l}(\mathcal{D}, \psi_0)$

**Input:** Training Data $\mathcal{D} := \{x^i, \theta^i\}_{i=1}^N$, Initial EBM parameters $\psi_0$
**Output:** Density estimator $q_\psi(x, \theta)$
**Initialize** $q_{\psi_0}(x) \propto e^{-E_{\psi_0}(x,\theta)}, \hat{q}_0 \propto \sum_i \delta_{(x^i, \theta^i)}$
**for** $k = 0, \ldots, K-1$ **do**
$\qquad \hat{q} := \texttt{make\_particle\_approx}(q_{\psi_k}, \hat{q})$
$\qquad \hat{G} = -\frac{1}{N} \sum \nabla_\psi E_{\psi_k}(x^i, \theta^i) + \mathbb{E}_{\hat{q}} \nabla_\psi E_{\psi_k}(x, \theta)$
$\qquad \psi_{k+1} = \texttt{ADAM}(\psi_k, \hat{G})$
**end for**
**Return** $q_{\psi_K}$

---

## 3.2 Targeted Inference using Sequential-UNLE

In this section, we introduce our second method, Sequential-UNLE (or SUNLE in short), which performs targeted inference for a specific observation $x_o$. SUNLE follows the traditional methodology of targeted inference by splitting the simulator budget $N$ over $R$ rounds (often equally), where in each round $r$, a likelihood estimate $q_{\psi_r^\star}(x|\theta)$ in the form of a conditional EBM is trained using all the currently available simulated data $\mathcal{D}$. This allows to construct a new posterior estimate $q_{\psi_r^\star}(\theta|x) = e^{-E_{\psi_r^\star}(x,\theta)} p(\theta)/Z(\psi_r^\star, \theta)$ which is used to sample parameters $\{\theta^{(i)}\}_{i=1}^{N/R}$ that are then provided to the simulator for generating new data $x^i \sim G(\theta^{(i)})$. The new data are added to the set $\mathcal{D}$ and are expected to be more similar to the observation of interest $x_o$. This procedure allows to focus the simulator budget on regions relevant to the single observed data of interest $x_o$, and, as such, is expected to be more efficient in terms of the simulator use than amortized methods (Lueckmann et al., 2021). Next, we discuss the learning procedure for the likelihood model and the posterior sampling.

**Learning the likelihood.** At each round $r$, the effective proposal $\pi$ of the training data available can be understood (provided the number of data points drawn at reach rounds is randomized) as a mixture probability: $\pi := \frac{1}{r}(\pi^{(0)}(\theta) + q_{\psi_1^\star}(\theta|x_o) + \ldots + q_{\psi_{r-1}^\star}(\theta|x_o))$ which is used to update the likelihood model. In this case, the analytical form of $\pi$ is unavailable as it requires computing the normalizing constants of the posterior estimates at each round, thus making the tilting approach introduced for AUNLE impractical in the sequential setting. Since currently available likelihood objectives (Kelly & Grathwohl, 2021; Du & Mordatch, 2019) for EBMs take as input *unconditional* (or *joint*) EBMs, a likelihood learning approach building on such objectives would require modeling and learning the entire joint distribution $\pi(\theta)p(x|\theta)$ including the proposal $\pi(\theta)$. This latter point is problematic since $\pi$ is not needed for inference, and can be highly complex (as it is set to be the current posterior estimate), increasing the difficulty of training. Instead, SUNLE learns a likelihood model maximizing the *average conditional log-likelihood*,

$$\mathcal{L}(\psi) = \frac{1}{N} \sum_{i=1}^N \log q_\psi(x^i|\theta^i), \quad \nabla_\psi \mathcal{L}(\psi) = -\frac{1}{N} \sum_{i=1}^N (\nabla_\psi E_\psi(x^i, \theta^i) + \overbrace{\mathbb{E}_{q_\psi(\cdot|\theta^i)} \nabla_\psi E_\psi(x, \theta^i)}^{\text{intractable}}) \quad (5)$$

where $(x^i, \theta^i)_{i=1}^N$ are the current samples. Unlike standard EBM objectives, this loss *directly* targets the likelihood $q_\psi(x|\theta)$, thus bypassing the need for modelling the proposal $\pi$. We propose in Algorithm 4 a method that optimizes this objective (previously used for normalizing flows in Papamakarios et al., 2019) when the density estimator is a conditional EBM. The intractable term of Equation (5) is an average over the EBM probabilities conditioned on all parameters from the training set, and thus differs from the intractable term of (1), composed of a single integral. Algorithm 4 approximates this term during training by keeping track of

one particle approximation $\widehat{q}_i = \delta_{\tilde{x}_i}$ per conditional density $q_\psi(\cdot|\theta^i)$ comprised of a single particle. The algorithm proceeds by updating only a batch of size $B$ of such particles using an MCMC update with target probability chain $q_{\psi_k}(\cdot|\theta^i)$, where $\psi_k$ is the EBM iterate at iteration $k$ of round $r$. Learning the likelihood using Algorithm 4 allows to use all the existing simulated data during training without re-learning the proposal, maximizing sample efficiency while minimizing learning complexity. The multi-round procedure of SUNLE is summarized in Algorithm 3.

**Posterior sampling.** Unlike AUNLE, SUNLE's likelihood estimate $q_\psi$ does not inherit the $(Z, \theta)$-uniformization property guaranteed by Proposition 1. As a consequence, its posterior $q_{\psi_R^\star}(\theta|x)$ is *doubly intractable* as it involves the intractable normalizing constant $Z(\psi_r^\star, \theta)$. Nevertheless, we propose to sample from $q_{\psi_R^\star}(\theta|x)$ using Doubly-Intractable MCMC techniques. We consider a custom robust doubly intractable implementation that allows for accurate inference even on challenging posteriors with no parameter tuning other than compute-related parameters like the number of warmup steps.

---

**Algorithm 3** Sequential-UNLE

**Input:** prior $p(\theta)$, simulator $G$, budget $N$, no. rounds $R$
**Output:** Posterior estimate $q_\psi(\theta|x)$
**Initialize** $\pi^{(0)} = p, \psi_0, q_{\psi_0,\pi} \propto e^{-E_{\psi_0}(x,\theta)}\pi(\theta)$
Get $\mathcal{D} = \{\theta^{(i)} \sim \pi(\theta), x^{(i)} \sim G(\theta, \cdot)\}_{i=1}^{N/R}$
**for** $r = 1, \ldots, R$ **do**
  Get $\psi_r^\star := \mathtt{maximize\_cebm\_log\_l}(\mathcal{D}, \psi_{r-1}^\star)$
  Set $\pi_{r+1}(\theta|x) := e^{-E_{\psi_r^\star}(x,\theta)}p(\theta)/Z(\psi_r^\star;\theta)$
  Get $\{\theta^i\}_{i=1}^{N/R} \sim \pi_{\psi_r^\star}$ via Doubly-Intr. MCMC
  Get $\mathcal{D} = \mathcal{D} \cup \{\theta^{(i)}, x^{(i)} \sim G(\theta^{(i)}, \cdot)\}_{i=1}^{N/R}$
**end for**
Infer using Doubly-Intr. MCMC on $q_{\psi_R^\star}(\theta|x)$

**Algorithm 4** $\mathtt{maximize\_cebm\_log\_l}(\mathcal{D}, \psi_0)$

**Input:** Training data $\mathcal{D} := \{\theta^{(i)}, x^{(i)}\}_{i=1}^N$, Initial EBM parameters $\psi_0$
**Output:** Cond. Density estimator $q_\psi(x|\theta)$
**Initialize** $q_{\psi_0} \propto e^{-E_{\psi_0}(\theta,x)}, \{\hat{q}_i = \delta_{x^i}\}_{i=1}^N$
**for** $k = 0, \ldots, K - 1$ **do**
  **for** $i = 0, \ldots, N - 1$ **do**
    $\widehat{q}_i := \mathtt{make\_particle\_approx}(q_{\psi_k}(\cdot, \theta^i), \widehat{q}_i)$
  **end for**
  $\widehat{G} = -\frac{1}{N}\sum \nabla_\psi E_{\psi_k}(x^i, \theta^i) + \mathbb{E}_{\widehat{q}_i} \nabla_\psi E_{\psi_k}(x^i, \theta^i)$
  $\psi_{k+1} = \mathtt{ADAM}(\psi_k, \widehat{G})$
**end for**
**Return** $q_{\psi_K}$

---

## 4 EXPERIMENTS

In this section, we study the performance and properties of AUNLE and SUNLE in three different settings: a toy model that highlights the failure modes of other synthetic likelihood methods, a series of benchmark datasets for SBI, and a real life neuroscience model.

**Experimental details** AUNLE and SUNLE are implemented using `jax` (Frostig et al., 2018). We approximate expectations of AUNLE's joint EBM using 1000 independent MCMC chains with a Langevin kernel parameterised by a step size $\sigma$, that automatically update their step size to maintain an acceptance rate of 0.5 during a per-iteration warmup period, before freezing the chain and computing a final particle approximation. Additionally, we introduce a new method which replaces the MCMC chains by a single Sequential Monte Carlo sampler (Chopin et al., 2020; Del Moral et al., 2006), which yields a similar performance as the Langevin-MCMC approach discussed above, but is more robust for lower computational budgets (see Appendix A.2). The particle approximations are persisted across iterations (Tieleman, 2008; Du & Mordatch, 2019) to reduce the risk of learning a "short run" EBM (Nijkamp et al., 2019; Xie et al., 2021) that would not approximate the true likelihood correctly (see Appendix B.2 for a detailed discussion). All experiments are averaged across 5 random seeds (and additionally 10 different observations $x_o$ for benchmark problems). We provide all code[1] needed to reproduce the experiments of the paper. Training and inference are computed using a single RTX5000 GPU. For benchmark models, a single round of EBM training takes around 2 minutes on a GPU (see Appendix B.4).

### 4.1 A TOY MODEL WITH A MULTI-MODAL LIKELIHOOD

First, we illustrate the issues that SNLE and SMNLE can face when applied to model certain distributions using a simulator with a bi-modal likelihood. Such a likelihood is known to be hard to model by normalizing flows, which, when fitted on multi-modal data, will assign

---

[1] https://github.com/anon-autors-a-sunle-iclr/unle

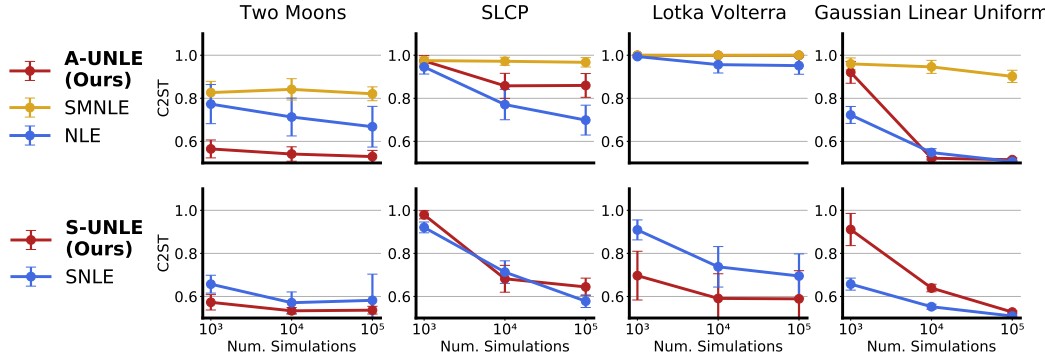

Figure 2: Performance of AUNLE (resp. SUNLE) compared with NLE and SMNLE (resp. SNLE), using the Classifier Accuracy Metric (Lueckmann et al., 2021) (lower is better). AUNLE and SUNLE exhibit robust performance across a wide array of problems. Additional details on the experimental setup can be found in Appendix B.5.

high-density values to low-density regions of the data in order to "connect" between the modes of the true likelihood (Cornish et al., 2020). Moreover, multi-modal distributions are also poorly handled by score-matching, since score-matching minimizes the Fisher Divergence between the model and the data distribution, a divergence which does not account for mode proportions (Wenliang & Kanagawa, 2020). Figure 1 shows the likelihood model learned by NLE and SMNLE on this simulator, which exhibit the pathologies mentioned above: the score-matched likelihood only recovers a single mode of the likelihood, while the flow-based likelihood has a distorted shape. In contrast, AUNLE estimates both the likelihood and the posterior accurately. This suggests that AUNLE has an advantage when working with more complex, possibly multi-modal, distributions, as we confirm later in Section 4.3.

## 4.2 RESULTS ON SBI BENCHMARK DATASETS

We next study the performance of AUNLE and SUNLE on 4 SBI benchmark datasets with well-defined likelihood and varying dimensionality and structure (Lueckmann et al., 2021):

**SLCP**: A toy SBI model introduced by (Papamakarios et al., 2019) with a unimodal gaussian likelihood $p(x|\theta)$. The dependence of $p(x|\theta)$ on $\theta$ is nonlinear, yielding a complex posterior.

**The Lotka-Volterra Model** (Lotka, 1920): An ecological model describing the evolution of the populations of two interacting species, usually referred to as preys and predators.

**Two Moons**: A famous 2-d toy model with posteriors comprised of two moon-shaped regions, and yet not solved completely by SBI methods.

**Gaussian Linear Uniform**: A simple gaussian generative model, with a 10-dimensional parameter space.

These models encompass a variety of posterior structures (see Appendix B.1 for posterior pairplots): the two-moons and SLCP posteriors are multimodal, include cutoffs, and exhibit sharp and narrow regions of high density, while posteriors of the Lotka-Volterra model place mass on a very small region of the prior support. We compare the performance of AUNLE and SUNLE with NLE and its sequential analogue SNLE, respectively: NLE and SNLE represent the gold standard of current synthetic likelihood methods, and perform particularly well on benchmark problems (Lueckmann et al., 2021). We use the same set of hyperparameters for all models, and use a 4-layer MLP with 50 hidden units and swish activations for the energy function. Results are shown in Figure 2.

While some fluctuations exist depending on the task considered, these results show that the performance of AUNLE (and SUNLE when targeted inference is necessary) is on par with that of (S)NLE, thus demonstrating that a generic method involving Energy-Based models can be trained robustly, without extensive hyperparameter tuning. Interestingly, the model where UNLE has the greatest advantage over NLE is Two Moons, which is the benchmark that exhibits a likelihood with the most complex geometry; in comparison,

the three remaining benchmarks have simple normal (or log-normal) likelihood, which are unimodal distributions for which normalizing flows are particularly well suited. This point underlines the benefits of using EBMs to fit challenging densities. Finally, we remark that SMNLE, which addresses only *amortized* inference Pacchiardi & Dutta (2022) struggled in practice for the toy problems investigated here.

### 4.3 USING SUNLE IN A REAL WORLD NEUROSCIENCE MODEL

We investigate further the performance of SUNLE by running its inference procedure on a simulator model of a pyloric network located in stomatogastric ganglion (STG) of the crab *Cancer borealis* given an observed an neuronal recording (Haddad & Marder). This model simulates 3 neurons, whose behaviors are governed by synapses and membrane conductances that act as simulator parameters $\theta$ of dimension 31. The simulated observations are composed of 15 summary statistics of the voltage traces produces by neurons of this network (Prinz et al., 2003; 2004). Amortized SBI methods require tens of millions of samples, while currently, the most sample-efficient targeted inference method for this problem is SNVI (a variant of SNLE that replaces the MCMC-powered posterior sampling by a variational inference step Glöckler et al. 2021) which uses 30 rounds simulating each 10000 samples.

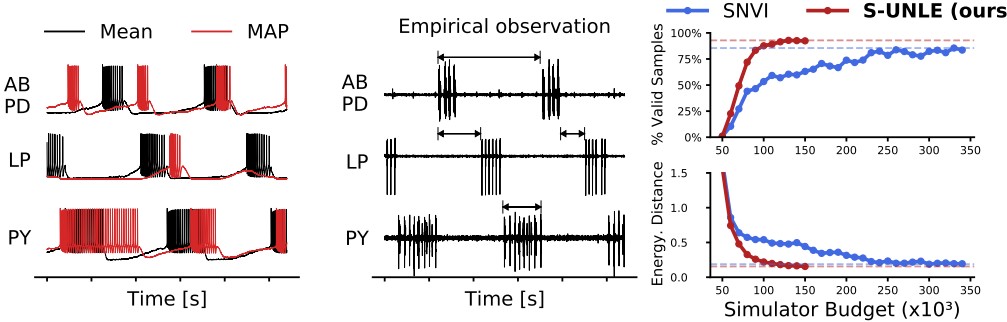

Figure 3: Inference with SUNLE on a model of the pyloric network. Left: simulations obtained by using the final posterior mean and maximum a posteriori (MAP) as a parameter. Center: the empirical observation $x_o$: arrows indicate the summary statistics. Top-right: fraction of simulated observations with well-defined summary statistics (higher is better) at each round for SNVI and SUNLE, with dashed lines indicating the maximum fraction for each method. Bottom-right: performance of the posterior using the Energy Distance.

We perform targeted inference on this model using SUNLE with a MLP of 9 layers and 300 hidden units per layers for the energy $E_\psi$, and perform doubly intractable MCMC to draw new proposal parameters across rounds. All inference and training steps are initialized using previously the available MCMC chains and EBM parameters. We report in Figure 3 the evolution of the rate of simulated obvservations with valid summary statistics, - a metric indicative of posterior quality - as well as the Energy-Scoring Rule (Gneiting & Raftery, 2007) of SUNLE and SNVI's posteriors across rounds. The synthetic observation simulated using SUNLE's posterior mean closely matches the empirical observation (Figure 3, Left vs Center). As shown in Figure 3, SUNLE matches the performance of SNVI in only 5 rounds, reducing by 6 the simulation budget of SNVI to achieve a comparable inference quality. After 10 rounds, SUNLE's poterior significantly exceeds the performance of SNVI in terms of number of valid samples obtained by taking the final posterior samples as parameters. The total procedure takes only 3 hours (half of which is spent simulating samples), *10 times less than SNVI.*

**Conclusion** The expanding range of applications of Simulation-Based Inference poses new challenges to the way SBI algorithms model data. In this work, we presented SBI methods that use an expressive Energy-Based Model as their inference engine, fitted using Maximum Likelihood. We demonstrated promising performance on synthetic benchmarks and on a real-world neuroscience model. In future work, we hope to see applications of this method to other fields where EBMs have been proven successful, such as physics (Noé et al., 2019) or protein modelling (Ingraham et al., 2018).

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

SUPPLEMENTARY MATERIAL FOR THE PAPER *Maximum Likelihood Learning of Energy-Based Models for Simulation-Based Inference*

The supplementary materials includes the following materials:

- A discussion of the computational rationale motivating the tilting approach of AUNLE.

- We propose a training method for EBM which uses the family of Sequential Monte Carlo samplers to efficiently approximate expectations under the EBM during approximate likelihood maximization. We show that using these new methods can lead to increased stability and performance for a fixed budget.

- An experiment in Appendix B.3 that suggest that the uniformization of AUNLE's posterior holds in learned AUNLE models.

- A discussion in Appendix B.2 about the (absence of) manifestation of the short-run effect [Nijkamp et al., 2019] in UNLE.

- A detailed computational analysis in Appendix B.4 of AUNLE, which proves highly competitive over alternatives.

- Figures in Appendix B.1 of UNLE's posterior sapmles for benchmark and the pyloric network's problems.

- Finally, we provide additional details in Appendix B.6 on the results of SUNLE on the pyloric network: we provide an estimation of the pairwise marginals of the final posterior, which contains patterns also present in the pairwise marginals obtained by Glöckler et al. [2021].

## A    METHOLOGICAL DETAILS

### A.1    ENERGY-BASED MODELS AS DOUBLY-INTRACTABLE JOINT ENERGY-BASED MODELS

AUNLE learns a likelihood model $q_\psi(x|\theta)$ by minimizing the likelihood of a tillted joint EBM $\frac{p(\theta)e^{-E_\psi(x,\theta)}}{Z_\pi(\psi)}$. While the gain in tractability arising in AUNLE's posterior suffices to motivate the use of this model, another computational argument holds. Consider the non-tilted joint model:

$$\pi(\theta)\frac{q_\psi(x|\theta)}{Z(\theta,\psi)}.$$

Expectations under this model can be computed by running a MCMC chain implementing a Metropolis-Within-Gibbs sampling method as in Kelly & Grathwohl [2021], which uses:

- any proposal distribution for $q_{\pi,\psi}(x|\theta) \propto q_\psi(x|\theta)$, such a MALA proposal

- an approximate doubly-intractable MCMC kernel step for $q_{\pi,\psi}(\theta|x) \propto \pi(\theta)\frac{e^{-E_\psi(x,\theta)}}{Z_\pi(\theta)}$ which is doubly intractable.

However, running the approximate doubly intractable MCMC kernel step requires sampling from $q_\psi(x|\theta)$, incurring an additional nested loop during training. Thus, naive MCMC-based Maximum-Likelihood optimization of untilted joint EBM is prohibitive from a computational point of view.

### A.2    TRAINING EBMs USING SEQUENTIAL MONTE CARLO

The main technique to compute particle approximations of the EBM iterations (returned by the generic `make_particle_approx`) when training an EBM using Algorithm 2 is to run N MCMC chains in parallel targeting the EBM Song & Kingma [2021]; aggregating the final samples $y_i$ of each chain $i$ yields a particle approximation $q = \frac{1}{N}\sum_i \delta y_i$ of the EBM in question. In this appendix section, we describe an alternative `make_ebm_approx` which

*efficiently* constructs EBM particle approximations across iterations of Algorithm 2 through a Sequential Monte Carlo (SMC) algorithm [Chopin et al., 2020; Del Moral et al., 2006]. In addition to its efficienty, this new routine does not suffer from the bias of incurred by the use of finitely many steps in MCMC-based methods. We apply this routine within the EBM training step of AUNLE's, and show that the learned posteriors can be more accurate than MCMC methods for a fixed compute power allocated to training.

### A.2.1 Background: Sequential Monte Carlo Samplers

Sequential Monte Carlo (SMC) Samplers [Chopin et al., 2020; Del Moral et al., 2006] are a family of efficient Importance Sampling (IS)-based algorithms, that address the same problem as the one of MCMC, namely computing a normalized particle approximation of a target density $q$ known up to a normalizing constant $Z$. The particle approximation $\widehat{q}_{SMC}$ computed by SMC samplers (consisting of $N$ *particles* $y^i$, like in MCMC methods, but weighted non-uniformly by some weights $w^i$) is produced by defining a set of $L$ intermediate densities $(\nu_l)_{l=0}^L$ bridging between the target density $\nu_l = q$ and some initial density $\nu_0$, for which a particle approximation $\nu_0^N : \sum_i = 1^N = w_0^i \delta_{y_0^i}$ are readily available. The intermediate densities are often chosen to be a geometric interpolation between $\nu_0$ and $\nu_L$, i.e. $\nu_l \propto (\nu_0)^{1-\frac{l}{L}} (\nu_l)^{\frac{l}{L}}$, so that $\nu_l$ are also known up to some normalizing constant. SMC samplers sequentially constructs an approximation $\nu_l^N := \sum w_l^i \delta_{y_l^i}$ to the respective density $\nu_l$ at time $l$, using previously computed approximations of $\nu_{l-1}$ at time $l-1$. At each time step, the approximations are obtained by applying three successive operations: *Importance Sampling*, *Resampling* and *MCMC* sampling. We provide a vanilla SMC sampler implementation in Algorithm 5, and refer to this algorithm as `make_smc_particle_approx`

---

**Algorithm 5** $\texttt{SMC}(q, \nu_0, \nu_0^N)$

---

1: **Hyper-parameters:** Number of particles $N$, number of steps $L$, re-sampling threshold $A \in [\frac{1}{N}, 1)$.
2: **Input:** Target density $q$, initial density $\nu_0$, particle approximations $\nu_0^N$ and $\nu_0$
3: **Output:** Particle approximations to $q$.
4: Construct geometric path $(\nu_l)_{l=1}^L$ from $\nu_0$ and $q$.
5: **for** $l = 1, \ldots, L$ **do**
6:     Compute IS weights $w_l^i$ and $W_l^i$
7:     Draw $N$ samples $(\widetilde{Y}_l^i)_{i=1}^N$ from $(Y_{l-1}^i)_{i=1}^N$ according to weights $(W_l^i)_{i=1}^N$, then set $W_l^i = \frac{1}{N}$.
8:     Sample $Y_l^i \sim \mathcal{K}_l(\widetilde{Y}_l^i, \cdot)$ using Markov kernel $\mathcal{K}_l$.
9: **end for**
10: Return approximation $q_{SMC}^N := (Y_L^i, W_L^i)_{i=1}^N$.

---

Importantly, under mild assumptions, the particle approximation constructed by SMC provides consistent estimates of expectations of any function $f$ under the target $q$:

$$\sum_{i=1}^N w^i f(y^i) \xrightarrow{P} \mathbb{E}_{y \sim q}[f(y)].$$

We briefly compare the role played by the number of steps and particles in both MCMC and SMC algorithms:

**Number of particles** SMC samplers differ from MCMC samplers in their origin of their bias: while the bias of MCMC methods comes from running the chain for a finite number of steps only, the bias of SMC methods comes from the use of finitely many particles.

**Number of steps** While it is usually beneficial to use a high number of iterations within MCMC samplers to decrease algorithm bias and ensure that the stationnary distribution is reached, the number of steps (or intermediate distributions) in SMC is beneficial to ensure a smooth transition from the proposal to the target distribution: however, the variance of SMC samplers as a function of the number of steps is not guaranteed to be decreasing

even if variance bounds that are uniform in the number of steps can be derived by making assumptions on $\mathcal{K}_l$ Chopin et al. [2020]. When applying SMC within AUNLE's training loop, we find that using more SMC samplers steps usually increase the quality of the final posterior.

In the next paragraph, we describe how to use SMC routine efficiently to approximate EBM expectations within Algorithm 2.

### A.2.2 EFFICIENT USE OF SMC DURING AUNLE TRAINING USING OG-SMC

A naive approach which uses the SMC routine of Algorithm 5 within the EBM training loop of Algorithm 2 would consist in calling the SMC at every training iteration using a fixed, predefined proposal density $\nu_0$ and associated particle approximation and $\hat{\nu}_0$, such as one from a standard gaussian distribution. However, as training goes, the EBM is likely to differ significantly from the proposal density $q_0$, requiring the use of many SMC inner steps to obtain a good particle approximation.

*A more efficient approach*, which we propose, is to use the readily available particle unnormalized EBM density $q_{\psi^{k-1}}$ and associated particle approximation $\hat{q}^k$ computed by SMC at the iteration k-1 **as the input** to the call to SMC targeting the EBM $q_{\psi^k}$ at iteration k. Algorithm 6 implements this approach.

---

**Algorithm 6** SMC-powered ML training of EBMs

---

**Input:** Training Data $\{x^{(i)}\}_{i=1}^N$, Initial EBM parameters $\psi_0$
**Output:** Density estimator $q_\psi(x)$
**Initialize** $q_{\psi_0}(x) \propto e^{-E_{\psi_0}(x)}, q_{-1} = \nu_0, \hat{q}_{-1} = \hat{\nu}_0$
**for** $i = 0, \ldots, \texttt{max\_iter} - 1$ **do**
    # $\hat{q} := \texttt{make\_particle\_approx}(q_{\psi_k}, \hat{q})$
    $\hat{q}_k := \texttt{SMC}(q_{\psi_k}, q_{k-1}, \hat{q}_{k-1})$
    $q_k := q_{\psi_k}$
    $\hat{G} = -\frac{\gamma}{N} \sum \nabla_\psi E_{\psi_k}(x^i) + \mathbb{E}_{\hat{q}} \nabla_\psi E(x)$
    $\psi_{k+1} = \texttt{ADAM}(\psi_k, \hat{G})$
**end for**
Return $q_{\psi_K}$

---

In practice, we find that using 20 SMC intermediate densities (with 3 steps of $\mathcal{K}_t$) in each call to SMC yields a similar performance as a 250-MCMC steps EBM training procedure. By considering a more constrained budget, using only 5 SMC intermediates densities outperforms a 30-steps MCMC EBM training procedure.

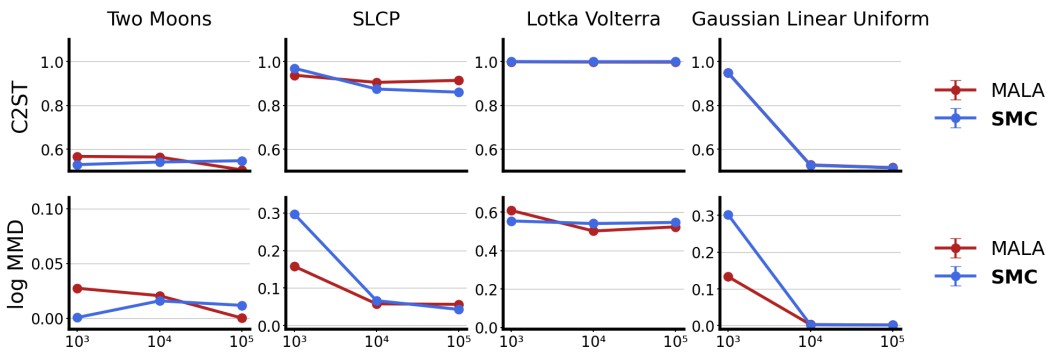

Figure 4: Performance of AUNLE, using either a MCMC-powered particle approximation routine, or a SMC: using 30 MCMC steps or 5 SMC steps

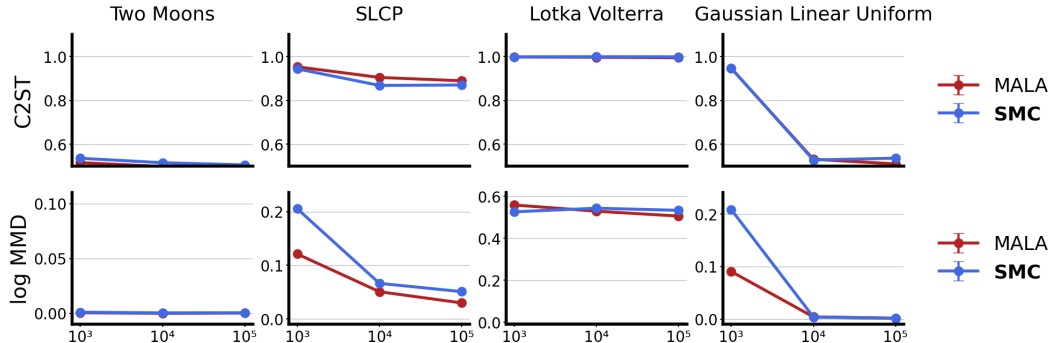

Figure 5: Performance of AUNLE, using either a MCMC-powered particle approximation routine, or a SMC: using 200 MCMC steps or 20 SMC steps

## B    Additional Experimental and Inferential Details

### B.1    Posterior pairplots on benchmark Problems

We report the ground truth estimated posterior pairplots on benchmark problems. AUNLE and SUNLE exhibit satisfying mode coverage, and are able to capture complex posterior structures.

### B.2    Manifestation of the short-run effect in UNLE

It was shown in Nijkamp et al. [2019] that EBMs trained by replacing the intractable expectation under the EBM with an expectation under a particle approximation obtained by running parallel runs of Langevin Dynamics initialized from random noised and updated for a fixed (and small) amount of steps can yield an EBM whose density is not proportional to the true density, but rather a generative model that can generate faithful images by running few steps of Langevin Dynamics from random noise on it. Our design choices for both training and inference purposefully avoid this effect from manifesting itself in UNLE. During training, we estimate the intractable expectation using *persistent* MCMC or SMC chains, e.g by initializing the MCMC (or SMC) algorithm of iteration $k$ with the result of the MCMC (or SMC) algorithm at iteration $k-1$, yielding a different training method than short-run EBMs. At inference, the posterior model is sampled from Markov Chains with a significant burn-in period, contrasting with the sampling model of short-run EBMs. Figure 7 compares the density of UNLE's posterior estimate for the two-moons model (a 2d posterior which can be easily visualized) with the true posterior. As the Figure 7 shows, AUNLE and SUNLE's posterior density match the ground truh very closely, demonstrating that UNLE's EBM is not a short-run generative model, but a faithful *density estimator*.

### B.3    Validating the $(Z, \theta)$ uniformization of AUNLE's posterior in practice

Proposition 1 ensures that the normalizing constant $Z(\theta; \psi)$ present AUNLE's posterior is independent of $\theta$ *provided that the problem is well-specified, and that $\psi = \psi^\star$*, the optimum of AUNLE's population objective. In practice, these conditions will hold exactly, and the uniformization of AUNLE's posterior thus only holds approximately. To assess the loss of precision associated with using a standard MCMC posterior in the context of approximate uniformization, we compare the quality of AUNLE's posterior samples obtained using a standard MCMC sampler (which is valid only uniformization holds), and using a doubly intractable MCMC sampler, which handles non-uniformized posteriors. We that approximation error of doubly intractable samplers by using a large number of steps (1000) when sampling from the likelihood using MCMC. As Figure 8 shows, there is no gain is using a doubly intractable sampler for inference in AUNLE, suggesting that the uniformization property of AUNLE holds well in practice.

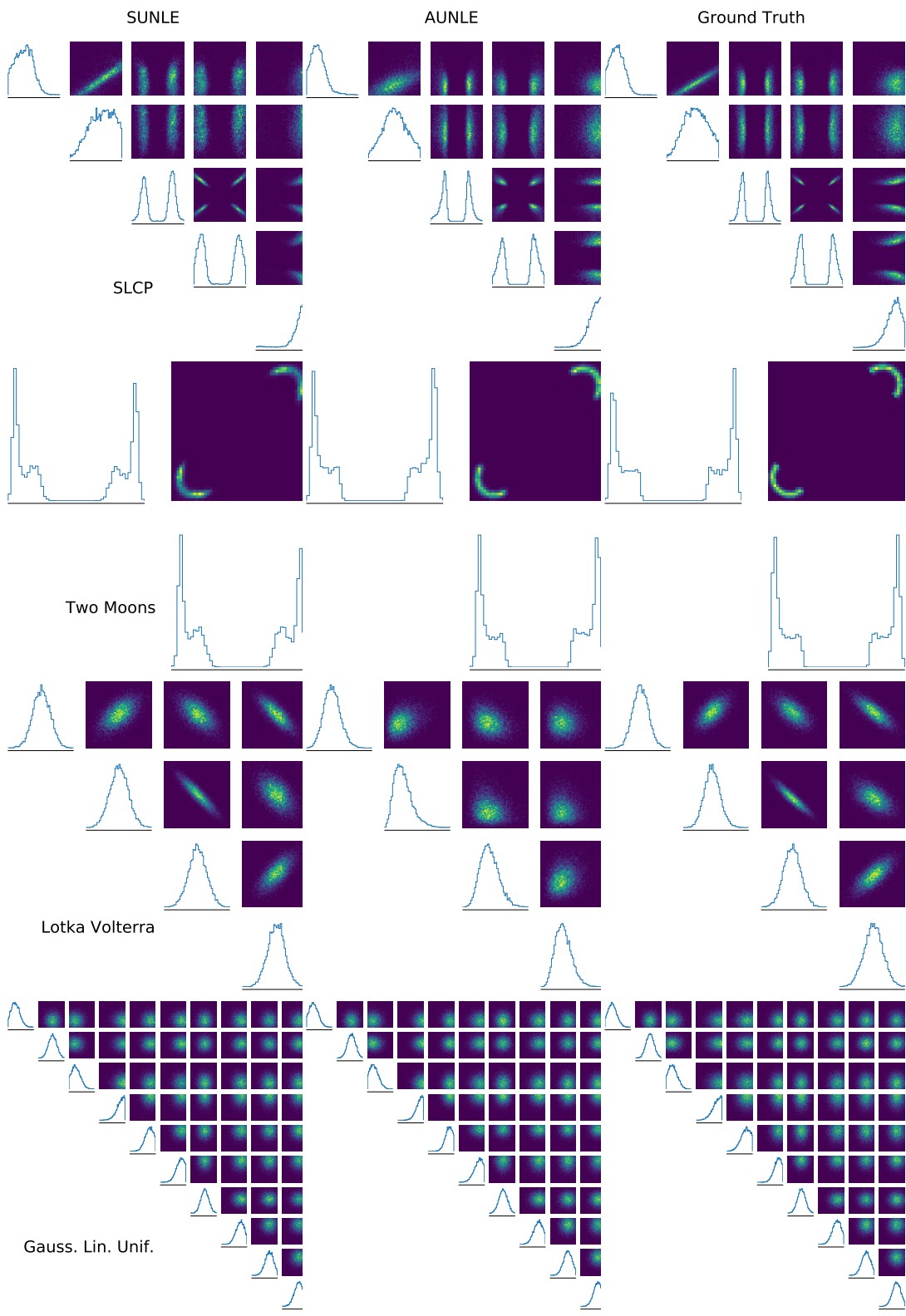

Figure 6: Posterior marginal (empirical) pairplots for SUNLE's posterior (first column), AUNLE's posterior (second column) and the ground truth posterior for the 4 studied benchmark problems. Each row outlines a separate benchmark problem.

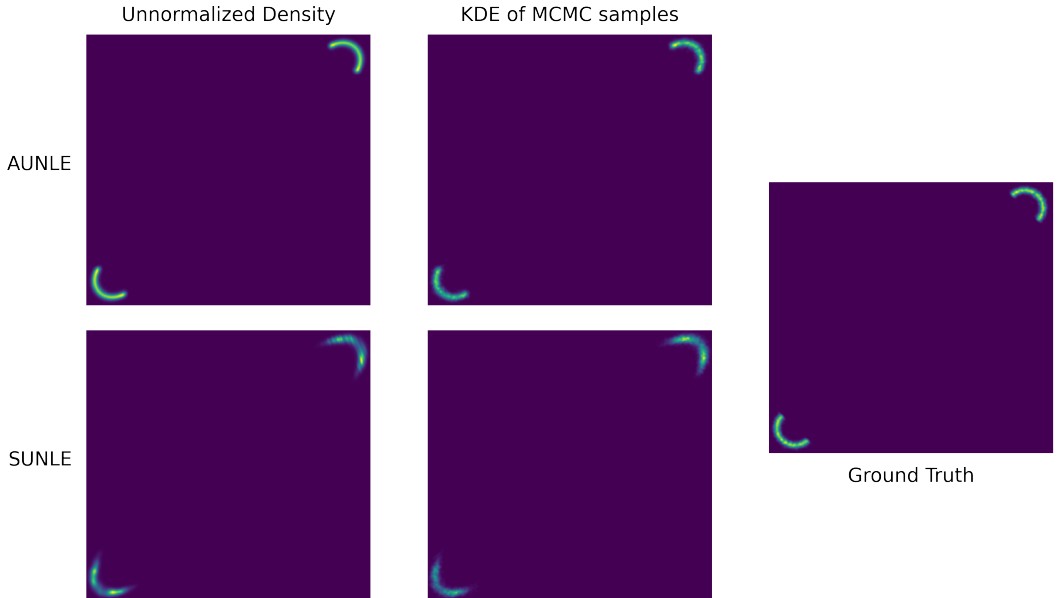

Figure 7: Normalized density of AUNLE and SUNLE for the two moons model. Left: manually normalized posterior density of both AUNLE and SUNLE using a discretization of the posterior over a grid. Middle: kernel density estimation of the MCMC samples obtained from AUNLE and SUNLE's posterior. Right: Ground Truth posterior. AUNLE and SUNLE's posterior densities match closely the true density, showing that these method indeed learn a density estimator, and a generative model [Nijkamp et al., 2019].

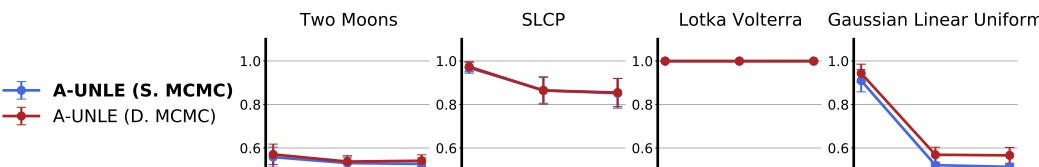

Figure 8: Quality of AUNLE's posterior samples (measured in classifier accuracy) obtained using a Standard MCMC sampler (S. MCMC) and a doubly intractable sampler (D. MCMC). The results show no gain in using a doubly intractable sampler, justifying the use of standard samplers for AUNLE.

### B.4 Computational Cost Analysis

Training unnormalized models using approximate likelihood is computationally intensive, as it requires running a sampler during training at each gradient step, yielding a computational cost of $O(T_1 T_2 N)$, where $T_1$ is the number of gradient steps, $T_2$ is the number of MCMC steps, and $N$ is the number of parallel chains used to estimate the gradient.

To maximize the efficiency of training, we implement all samplers using jax Frostig et al. [2018], which provides a Just–In–Time compiler and an auto-vectorization primitive that generates efficient, custom parallel sampling routines. For AUNLE, we the introduced a warm-started SMC approximation procedure to estimate gradients, yielding a competitive performance with as little as 5 intermediate probabilities per gradient computation. For SUNLE, we warm-start the parameters of the EBM across training rounds, and warm-start the chains of the Doubly Intractable sampler accross inference rounds, which significantly reduces the need for burning steps and long training. Finally, all experiments are ran on GPUs. Together, these techniques make AUNLE and SUNLE almost always the fastest methods for amortized and sequential inference, with total per-problem runtimes of 2 for AUNLE and 15 minutes for SUNLE on benchmark models (which is significantly faster than NLE and SNLE on their canonical CPU setup Lueckmann et al. 2021) and less than 3 hours for SUNLE on the pyloric network model (with half of this time spent simulating samples). The latter is *10 times faster than SNVI* (30 hours) on the same model. A breakdown of training, simulation and inference time is provided in Figure 9. We note that (S)NLE was ran on a CPU, which is the advertised computational setting [Lueckmann et al., 2021], since (S)NLE deep and shallow networks that do not benefit much from GPU acceleration.

We note that the time spent performing inference is negligible for AUNLE, which uses standard MCMC for inference thanks to the tilting trick employed in its model. On the other hand, the runtime of SUNLE, which performs inference using a doubly intractable sampler is dominated by its inference phase. This point demonstrates the computational benefits of the AUNLE's tilting trick. Note that SUNLE performs inference in a multi-round procedure, and requires thus $R$ training and inference phases (where $R$ is the number of rounds), as opposed to 1 for AUNLE. We alleviate this effect by leveraging efficient warm-starting strategies for both training and inference, which allow to amortize these steps across rounds.

### B.5 Experimental setup for SNLE and SMNLE

**SNLE**   The results reported for SNLE are the one present in the SBI benchmark suite [Lueckmann et al., 2021], which reports the performance of both NLE and SNLE on all benchmark problems studied in this paper.

**SMNLE**   The results reported for SMNLE were obtained by running the implementation referenced by Pacchiardi & Dutta [2022]. SMNLE comes in two variants: the first variant uses plain Score Matching [Hyvärinen & Dayan, 2005] to estimate its conditional EBM, while the second variant uses Sliced Score Matching [Song et al., 2020], which yields significant computational speedups during training. For both methods, we train the model using 500 epochs, and neural networks with 4 hidden layers and 50 hidden and outputs units. To optimize the inference performance, we carry out inference using our own Doubly Intractable Sampler, which automatically tunes the all parameters of the doubly intractable samplers except for the number of burn-in steps, and initializes the chain at local posterior modes. We carry out a grid search over the learning rates 0.01 and 0.001, and leave other training parameters to their default. Figures in the main body only report the performance of the Sliced Score Matching variant, which perform better in practice and runs order of magnitude faster. Figure 10 reports the performance of both variants for completeness. We used GPU both to train and inference using SMNLE, yielding similar or higher training compared to AUNLE for the sliced variant, and much longer training times for the standard variant.

### B.6 Neuroscience Model: Details
**Pairwise Marginals**   We provide the full pairwise marginals obtained after computing a kernel density estimation on the final posterior samples of SUNLE. We retrieve similar

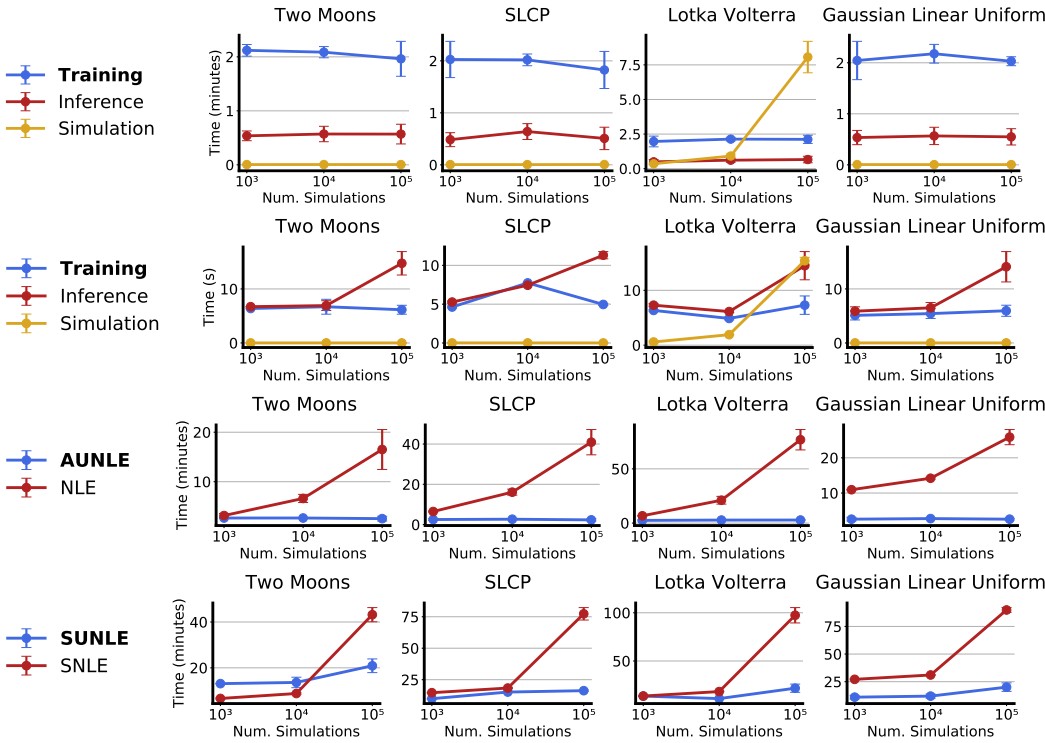

Figure 9: Runtime of UNLE: Analysis and Comparisons.
First row: time (in minutes) spent training, inferring, and simulating for AUNLE. Second row. Second row: time (in minutes) spent training, inferring, and simulating for SUNLE. Third row: runtime comparison between ANLE and NLE (in log-scale). Forth row: runtime comparison between SUNLE and SNLE.

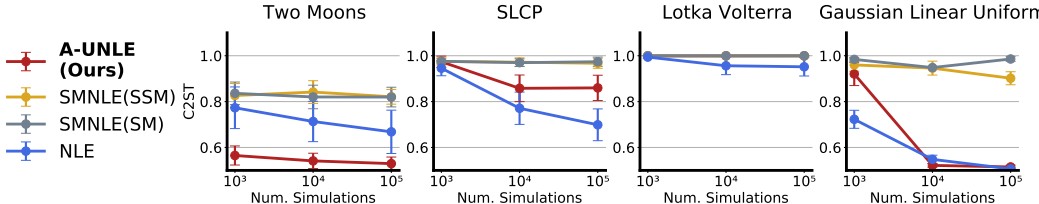

Figure 10: Comparison of AUNLE, SMNLE with Sliced Score Matching (SSM), SMNLE with Score Matching (SM) and NLE on a set of benchmark problems.

patterns as the one displayed in the pairwise marginals of SNVI samples. We refer to Glöckler et al. [2021] for more details on the specificities of this model.

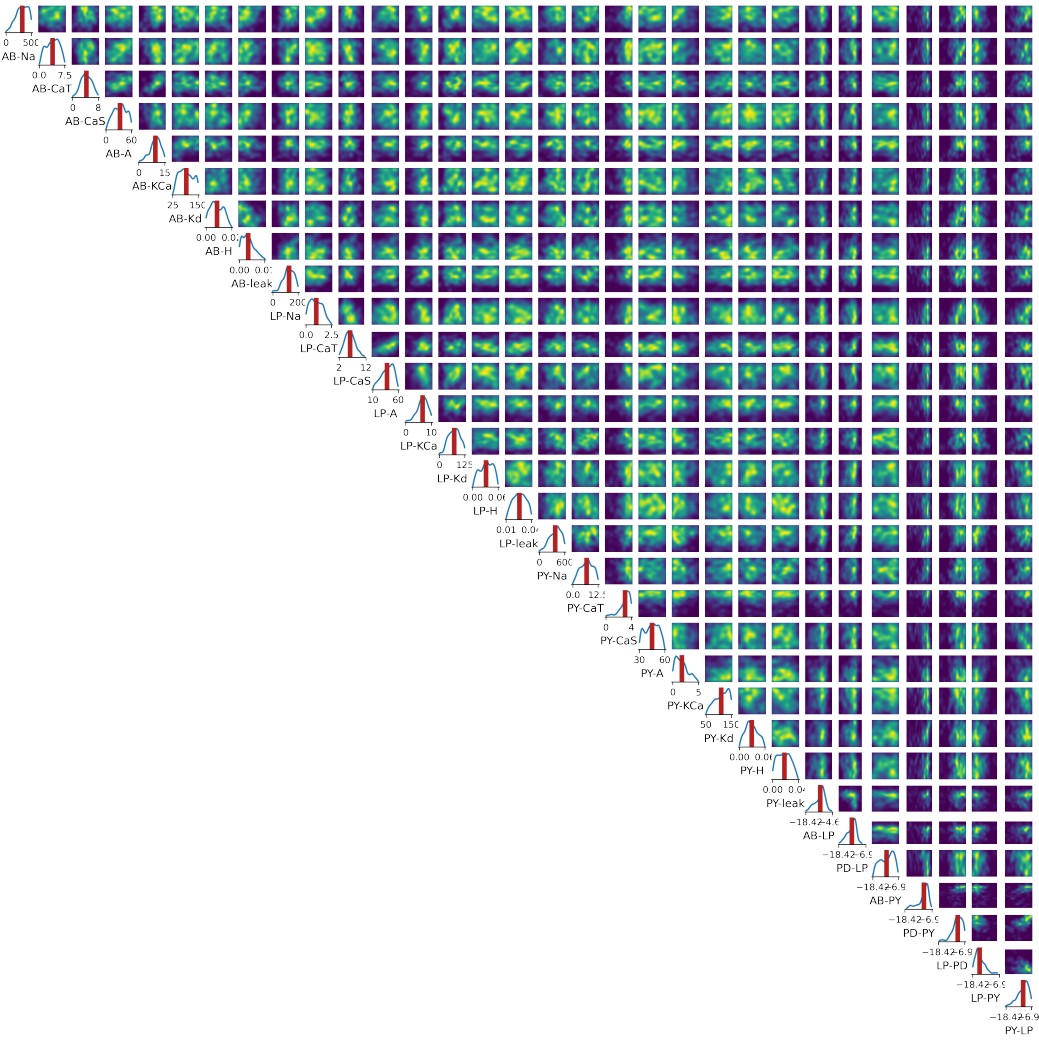

Figure 11: Pairwise marginals of SUNLE's posterior estimate on the *C. borealis* simulator model.

**Use of a Calibration Network**  Due to the presence of invalid observations, we proceed as in Glöckler et al. [2021] and fit a calibration network that allows to remove the bias induced by throwing away pairs of (parameters, observations) when the observations do not have well defined summary statistics. We use a similar architecture as in Glöckler et al. [2021].

