# OpenReview forum: "Maximum Likelihood Learning of Energy-Based Models for Simulation-Based Inference"
_ICLR.cc/2023/Conference — Submitted to ICLR 2023_

### Official Review · Reviewer_kgMn · 2022-10-25

**Confidence:** 4
**Correctness:** 3
**Technical Novelty And Significance:** 2
**Empirical Novelty And Significance:** 2
**Recommendation:** 5

**Clarity, Quality, Novelty And Reproducibility:**

The AUNLE method deals with a major theoretical limitation of existing SBI methods and is the most technically original part of the work. The SUNLE method is a straightforward application of EBM learning to a conditional density estimation problem that was previously tackle by flows and score models. More thorough experimental results would improve the paper, although I am not familiar with the area of SBI and I cannot suggest specific examples.

**Strength And Weaknesses:**

STRENGTHS

1. The AUNLE formulation provides a nice way of dealing with a major limitation of double intractibility by tilting the normalizer. There are clear methodological and computational benefits to this approach, but it still remains limited in its ability to fine-tune the learned posterior to a specific observed datapoint.
2. Experimental results show that AUNLE and SUNLE can outperform score-based SBI for multimodal datasets, and that SUNLE can outperform score-based SBI for complex biological data.

WEAKNESSES

1. The presentation of equations and algorithms is not always clear. In Algorithm 2, should the dataset be $(X_i, \theta_i)$ instead of just $(X_i)$? In Equation 5 and Algorithm 4 is the data variable in the intractible term meant to have the index on the $X_i$, or should it be $X$ like in Algorithm 2, since it is a generated sample and not a data sample?
2. The novelty of the SUNLE method is somewhat limited because it simply accepts the doubly intractible limitation and proceeds with standard EBM learning. Standard methods for flow and score models have already been used for a nearly identical purpose.
3. It would be interesting to see other real-world applications similar to the crab synapses, although benchmarking and evaluation are likely difficult.

**Summary Of The Paper:**

This work proposes new methods for Simulation Based Inference. The goal is to perform Bayesian inference on model parameters given data for situations where the likelihood function is not known in closed form and can only be accessed through a simulation mechanism. The central challenge of SBI learning is "doubly intractibility" because the normalizer of the posterior distribution depends on the model parameter. This work proposes two energy-based methods for SBI. The first bypasses the doubly intractibility by defining a joint EBM over $X$ and $\theta$ so that the normalizer does not depend on $\theta$. This allows amortized posterior sampling for any observed data. To tailor the posterior inference to a specific observed datapoint, the work further proposes to use an EBM to learn the conditional distribution and performs doubly intractible inference. Results show competitive performance with score based SBI models for toy datasets and a crab biology dataset.

**Summary Of The Review:**

The most interesting part of the paper is the tilting of the normalizer in the AUNLE method, although unfortunately this method likely will not be able to achieve the performance of models tailored towards inference for specific data. The lack of novelty of the main SUNLE method and sparse experimental results lead me to recommend not accepting this paper at this time.

---

> ### Author Response · Authors · 2022-11-18
> **Response to reviewer - thank you for your review**
>
> We thank the reviewer for their review. We address the reviewer's concerns below.
>
> **About the limitation of AUNLE:** It is true that AUNLE learns a posterior that is not tied to a specific observation. However, this aspect of AUNLE can be seen as a feature and not a limitation, since one need not to re-train AUNLE each time inference has to be performed on a new observation: this is the setting of *amortized inference*, which differs from the *targeted inference* setting (where inference is tied to a specific $x_o$).
>
> **About the equations and algorithms:**
> - Algorithm 2 describes an *unconditional* EBM learning procedure for generic inputs $X$, and can be specialized to the case when $X$ is a joint sample $(x, \theta)$. We agree that using $X$ to denote a generic random variable was confusing in this context. Our updated manuscript specializes this algorithm to joint samples $(x, \theta)$.
> - Equation 5 indeed contains a typo: $X_i$ should just be $X$. We thank the reviewer for raising that point to our attention.
>
> **About the novelty of SUNLE:** We would like to underline a number of distinctions between our modeling and training procedure for SUNLE, and the existing EBM approaches (LeCun et al., 2006; Du & Mordatch, 2019; Nijkamp et al., 2019; Kelly & Grathwohl, 2021):
>
> 1. Modeling: training a likelihood model using existing approaches would require extending the likelihood model into a joint  (unconditional) energy-based model $p(x, \theta)$, and maximize the likelihood of the extended model (Kelly et. Al, 2021). In the SBI setting, this approach would target the true joint  distribution $p(x|\theta)\pi(\theta)$ and would make the EBM re-learn the proposal $\pi(\theta)$ during training, which is not needed for inference. This relearning step may thus needlessly increase the complexity of  training, especially if the proposal is complex (as often the case in sequential inference, where $\pi$ is the current posterior estimate), or of higher dimension than the space of observations. Unlike such approaches, SUNLE learns a conditional density model $q_\psi(x|\theta)$ of the *likelihood* $p(x|\theta)$ is optimized using a loss *directly targeting the likelihood model* $q_\psi(x|\theta)$, and does not re-learn the prior during training.
> 2. Training: Second, although our training method shares similarities with training EBM with CD, the difference between SUNLE's objective and the maximum likelihood objective of standard EBMs makes SUNLE's algorithm to depart from standard Contrastive Divergence: for instance, the intractable term in SUNLE's objective must be approximated using separate MCMC chains targeting $q_\psi(x|\theta^i)$ for training parameters $\theta^i$, which is unlike the setting of traditional EBMs. We have updated the section 3.2 of the paper dedicated to SUNLE in order to highlight better the novelty and benefits of SUNLE's conditional EBM training method.
>
> Our updated manuscript clarifies the novelty and benefits of SUNLE.
>
>
> **About other real-world applications:** Our experiments have two objectives:
> 1. Illustrate the behavior of our method in well-understood environments
> 2. Apply our method to a challenging real-world example.
>
> For the first objective, we indeed followed a widely used experimental setup in the SBI literature: *the SBI benchmark suite* (Lueckmann et. al.; 2021), a benchmark suite with challenging posteriors (see Appendix B.1). We emphasize that many milestone SBI publications like  NLE (Papamakarios et al., 2019), NPE (Greenberg et al., 2019) reported results on benchmark models as they are well understood, and allow for a fair comparison between methods.
>
> For the second objective, we used *a real-world example on a full-scale simulator used in computational neuroscience*. The real-world experiment tackles a high-impact problem, with numerous highly cited works (Prinz et al., 2003; 2004; Haddad & Marder) dedicated to solving inference and simulation for this model already published. Inference for this model is challenging: only a narrow region (1%) of the prior support yields valid neural dynamics, with state of the art methods achieving only 85% of such dynamics on final posterior samples. Our proposed method significantly improves the quality of inference upon prior methods, and we thus believe that our method could benefit domain scientists for inference tasks with complex likelihoods in a domain where normalizing flows constitute the dominant modeling choice.
>
> We agree with the reviewer that more experiments are always useful. However, we believe that the significant improvement in performance of SUNLE in a highly studied problem demonstrates well the potential of UNLE to help domain scientists to perform more precise inference.
>
> ***
>
> We hope we have addressed your concerns in the above - if so, we would be grateful if you could consider raising the review score. If any questions or concerns remain, then please let us know.

---

> > ### Comment · Reviewer_kgMn · 2022-12-06
> > **Thanks for the responses**
> >
> > Thanks to the authors for their responses. Here are my thoughts on comment points raised by myself and other reviewers:
> >
> > 1. I still don't see how SUNLE is different from standard conditional EBM learning, such as the conditional ImageNet generation experiment from Du et al. 2019. In conditional EBM learning with image datasets, one wouldn't learn the prior over the condition distribution. I don't think this should be highlighted as a novel technique.
> > 2. The short-run effects might not be fully investigated. Although I did not bring this up in my review, it is an important point. From my experience, the short-run effect tends to manifest itself mostly strongly in the high dimensional case and well-formed densities in low-dimensional cases such as Figure 7 are not evidence of proper density learning in high dimensions. Furthermore, from my own experience and reports from prior work such as [1], persistent initialization is not sufficient to overcome the shortrun effect without careful tuning and implementation. Since the SBI benchmarks are low-dimensional, I find it plausible that the short-run effect will not manifest, but I would still be cautious to claim that the method totally avoids the short-run effect. To properly evaluate the short-run effect, the authors might consider an ablation study for the 31-dimensional crab synapse experiment that reports their results over a variety of MCMC trajectory lengths (say 10 steps, 100, steps, 1000 steps, 10000 steps, 100000 steps).
> > 3. The uniformization point about AUNLE raised by reviewer m5HY is subtle but quite important and I am glad to see that it has been addressed in the revision.
> >
> > For me, the deciding factor for this work is the quality of the experimental results. I am not an expert in SBI so I cannot reliably attest to the quality of the results, but the work claims to achieve significantly better parameter inference than existing methods at a much faster speed. This is the main appeal of the proposed method beyond the interesting idea of AUNLE. Nonetheless, the experimental settings are quite limited and for the most part low-dimensional (although this is not necessarily the fault of the authors). I will have decided to slightly raise my score.
> >
> > [1] On the anatomy of mcmc-based maximum likelihood learning with energy-based models. https://arxiv.org/pdf/1903.12370.pdf

---

> > > ### Author Response · Authors · 2022-12-09
> > > **Response to reviewer**
> > >
> > > We thank the reviewer for their additional suggestions, and for updating their score. We address the reviewer's remarks below:
> > >
> > > **About conditional training in [2]:** We thank the reviewer for pointing us to experiments described in [2], which, after careful inspection of the accompanying [code](https://github.com/openai/ebm_code_release/) indeed adopts a similar conditional training approach as ours. However, [2] described neither an objective function for training conditional EBMs, nor a way to approximate this objective's gradient, while we elucidate both these points in our paper. Also, while [2] considered only conditioning over discrete-valued variables, we show that it is possible to train conditional EBM with continuously-valued conditioned variables. We will make both points clearer in the final version of the manuscript.
> > >
> > >
> > > **About the short-run effect** We agree with the reviewer that it is possible that the short run effect manifests itself in the case of very high-dimensinonal models; however, many applications of SBI involve require only dealing with variables of moderate dimension thanks to expert knowledge. But we agree that investigating short-time effect of MCMC in the context of SBI would be an interesting direction for future work. We will also add the proposed ablation in our revised paper. We would like to point out to the reviewers that our training approach follows the recommendations of [1] for long-run EBMs: In particular, we use a consistent MCMC sampler (akin to setting $\tau=1$ in [1] and adding a Metropolis-Hasting correction step), we adapt the MCMC proposal step size $\epsilon$ **in an adaptive manner during training**, and we preform persistent initialization. We will make these points clearer in the paper.
> > >
> > >
> > > **About the experiments** Simulation-Based Inference aims at performing inference in domain sciences, like neuroscience, ecology or particle physics. In such problems, the random variables of interest can be high dimensional, but are often reduced to lower-dimensional variables thanks to summary statistics crafted by domain experts: this is on the said lower-dimensional statistics that SBI models are trained on. The SBI setting thus differs from traditional Computer Vision or NLP settings.
> > >
> > > [1] On the anatomy of MCMC-based maximum likelihood learning with energy-based models. https://arxiv.org/pdf/1903.12370.pdf
> > >
> > > [2] Implicit Generation and Modeling with Energy-Based Models, https://arxiv.org/pdf/1903.08689.pdf

---

### Official Review · Reviewer_m5HY · 2022-10-25

**Confidence:** 4
**Correctness:** 3
**Technical Novelty And Significance:** 3
**Empirical Novelty And Significance:** 3
**Recommendation:** 8

**Clarity, Quality, Novelty And Reproducibility:**

The approach proposed in this paper is, to my knowledge, novel.  It is interesting, and clearly presented.  The authors have code for reproducing results available at an anonymous github repo.  I did not run their code, but it looks thoroughly documented.

**Strength And Weaknesses:**

Strengths
* The proposed approach is straightforward, conceptually neat, and very interesting.
* The paper is well-written and the method and results and presented exceptionally clearly.
* I appreciate that in the toy models the strengths _and_ weaknesses of the present approach relative to existing methods (e.g., normalizing flows) are presented.  I think that the gains of the present approach on complex posteriors are sufficiently interesting (and well-explained and well-motivated) to more than outweigh the comparable (or slightly worse) performance to existing methods on simpler toy problems.

Weaknesses:
* The key assumption of AUNLE is $(Z, \theta)$-uniformization.  This allows one to interpret the learned energy model as a likelihood up to an unknown normalizing constant that is independent of $\theta$.  If this uniformization does not hold, then the normalizing constant will depend on $\theta$, resulting a double-intractable problem.  Given that uniformization only holds if 1) the energy model is sufficiently flexible such that the true joint distribution lies within the space of learnable models; and 2) the optimization problem can be exactly, globally solved, it seems like in practice uniformization will always be violated.  Such assumptions are frequently made in SBI (e.g., it is common to plug approximate posteriors into various formulas involving the true posterior) so I don't think that this is deeply problematic (and indeed such assumptions are often justified by good empirical performance), but I would be interested to see some empirical results about how much uniformization is violated in practice and how much that affects downstream inference.  One possibility would be to run double-intractable methods on the learned energy model (without assuming that the normalization is independent of $\theta$) for an extremely long time to obtain a "ground truth" and compare to the output of AUNLE.  Another possibility would be to estimate different normalizing constants for the implied likelihoods for different values of $\theta$, and see the extent to which they differ.  In any case, I think it should be made more explicit in the main text that this uniformization is unlikely to _exactly_ hold in practice, and the rest of the AUNLE methodology hinges on it holding  _approximately_.
* For all results it would be good to show some degree of replicability across initializations and random seeds.  How different are the different methods relative to differences in neural network initialization and randomness in the simulations?  Are the exact same simulations used to train the different methods, or are different random sets used?  E.g., the difference between using MALA and SMC within AUNLE results in comparable differences on SLCP as compared to the difference between AUNLE and NLE.

Typos:
* "making naive Bayesian inference impossible" --> "making standard Bayesian inference impossible"
* "each of which endowed with" --> "each of which is endowed with"
* Species names are never capitalized: "_Cancer Borealis_" --> "_Cancer borealis_"
* "given an observed an neuronal recording" --> "given an observed neuronal recording"
* "We show that using this new methods can" --> "We show that using these new methods can"
* "does not suffer from the bias of incurred by the" --> "does not suffer from the bias incurred by the"
* "the number of steps (or intermediate distributions) is SMC is beneficial" --> " the number of steps (or intermediate distributions) in SMC is beneficial"
* "When applying SMC to within AUNLE’s training" --> "When applying SMC within AUNLE’s training"
* "using more SMC samplers steps usually increase the quality" --> "using more SMC samplers steps usually increasea the quality"
* "comparing multirond sequential methods" --> "comparing multiround sequential methods"
* "posterior estimate on the _C. Borealis_ simulator" --> "posterior estimate on the _C. borealis_ simulator"

**Summary Of The Paper:**

This paper proposes a method for learning synthetic likelihoods in the likelihood-free setting of simulation-based-inference (SBI).  The method fits a particular tilted version of the joint distribution of the parameters and observations.  Under certain assumptions, the solution of this optimization problem is proportional to the conditional density of the data given the parameters.  As such, the authors then use this unnormalized conditional density to estimate posteriors using standard approaches (e.g., MCMC).  The authors also present a sequential version of this approach.  The authors apply these methods to 4 toy datasets and compare to other SBI posterior inference methods, where they obtain better performance on one of the toy problems.  Finally, they apply their method to a neuroscience dataset, where they can obtain a good approximate posterior using fewer simulations than a previous method.

**Summary Of The Review:**

Overall, I think this is a well-written interesting paper providing a promising approach for simulation-based-inference.

---

> ### Author Response · Authors · 2022-11-18
> **Response to reviewer - thank you for your review**
>
> We thank the reviewer for their thorough review, positive feedback and comments.  We address the reviewer's concerns below:
>
> **About the $(Z, \theta)$-uniformization** We thank the reviewer for raising that point, and suggesting ways to study it. Our updated manuscript now mentions explicitly the fact that $(Z, \theta)$-uniformization only holds approximately in practice, and points to an additional investigation (present in Appendix B.3) on how well this property is satisfied in practice. The investigation follows the suggested methodology of running a Doubly Intractable sampler for a large number of inner sampler steps (1000, 10 times more than during inference in SUNLE), and comparing the quality of inference between the resulting samples and the samples obtained using standard MCMC. The investigation shows little difference in performance between the two cases, suggesting that uniformization holds well in practice.
>
>
> **About the replicability of experiments across seeds** We thank the reviewer for raising that point. The figures of our updated manuscript now show the results for experiments *averaged across  5 seeds (as well as the original 10 observations for the benchmark problems)*, both for the benchmark problems and the real world problem. We note little variability across seeds, in particular for the real-world example, highlighting the robustness of our method in practice.
>
> **About the difference in simulations**, AUNLE-MCMC and AUNLE-SMC are trained using the same simulations, while AUNLE and NLE are not.
>
> Finally, we thank the reviewer for pointing out the typos presnet in our manuscript, which we have fixed.

---

> > ### Comment · Reviewer_m5HY · 2022-12-06
> > **Thank you for the response**
> >
> > Thank you for responding and performing these additional experiments.

---

### Official Review · Reviewer_jDWH · 2022-10-27

**Confidence:** 4
**Clarity, Quality, Novelty And Reproducibility:** The paper is well-written easy to fol…
**Correctness:** 4
**Technical Novelty And Significance:** 3
**Empirical Novelty And Significance:** 2
**Recommendation:** 5

**Strength And Weaknesses:**

AUNLE is well-motivated, the approach is interesting and novel, and its correctness has been backed up by Proposition 1.
However, SUNLE is simply training EBMs with CD, I am not seeing any novelty there.

I don't understand why the authors didn't use SMNLE as a baseline. They ruled out SMNLE for being double-intractable, but they are already using double-intractable MCMC for their real-world setup. The comparison in Figure 1 is not enough to rule out SMNLE from the baselines. Moreover, SMNLE shows a better likelihood ratio than NLE in Figure 1.


**Summary Of The Paper:**

The paper introduces (adapts) ML training of EBMs for SBI problems.
The main problem in SBI is that one cares about sampling from posterior $q(\theta|x)$, and in the case of using EBMs, sampling from posterior results in doubly intractable inference since the partition function depends on $\theta$ as well.
The paper suggest parameterizing the joint model $q_\psi(\theta, x)$ as $\pi(\theta)\exp(-E\psi(\theta, x))/Z_\psi$ and show that the optimal posterior also have similar form:  $q(\theta | x) = p(\theta)\exp(-E\psi(\theta, x))/Z_\psi$. This is important since they now can directly sample from the posterior without worrying about the partition function. The main limitation of this approach is that they need to have an analytical form p(\theta) as prior, which is not always available, especially for the sequential setup, in which the prior is defined using posteriors on the previous rounds. Therefore, the sequential version of their approach (SUNLE) just trains an EBM using contrastive divergence and uses doubly-intractable MCMC for inference.

The authors compare AUNLE and SUNLE with NLE and SNLE, which use normalizing flow, on four toy datasets and one real-world example.


**Summary Of The Review:**

The paper is well-motivated and discusses an important application. However, the main novelty of the paper is the introduction to AUNLE, however, the amortized version is not very useful in practice since using sequential modeling reduces the number of required samples.
I am not seeing any novelty in the sequential version of their work except only training the EBM using CD in their setup.

---

> ### Author Response · Authors · 2022-11-18
> **Response to reviewer - thank you for your review**
>
> We thank the reviewer for their comments on our submission. We address the reviewer's comments below:
>
> **About the novelty of SUNLE**: We would like to underline a number of distinctions between our modeling and training procedure for SUNLE, and the existing EBM approaches (LeCun et al., 2006; Du & Mordatch, 2019; Nijkamp et al., 2019; Kelly & Grathwohl, 2021 )
>
> 1. Modeling: training a likelihood model using existing approaches (Kelly et. Al, 2021; Du et. Al, 2019) would require extending the likelihood model into a joint  (unconditional) energy-based model $q_\psi(x, \theta)$, and maximize the likelihood of the extended model. In the SBI setting, this approach would target the true joint  distribution $p(x|\theta)\pi(\theta)$ and would make the EBM re-learn the proposal $\pi(\theta)$ during training, which is not needed for inference. This relearning step may thus needlessly increase the complexity of  training, especially if the proposal is complex (as often the case in sequential inference, where $\pi$ is the current posterior estimate), or of higher dimension than the space of observations. Unlike such approaches, SUNLE learns a conditional density model $q_\psi(x|\theta)$ of the *likelihood* $p(x|\theta)$ is optimized using a loss *directly targeting the likelihood model* $q_\psi(x|\theta)$, and does not re-learn the prior during training.
> 2. Training: Second, although our training method shares similarities with training EBM with CD, the difference between SUNLE's objective and the maximum likelihood objective of standard EBMs makes SUNLE's algorithm depart from standard Contrastive Divergence: for instance, the intractable term in SUNLE's objective is approximated using separate MCMC chains targeting $q_\psi(x|\theta^i)$ for training parameters $\theta^i$, which is unlike the setting of traditional EBMs. We have updated the section 3.2 of the paper dedicated to SUNLE in order to highlight better the novelty and benefits of SUNLE's conditional EBM training method.
>
> Our updated manuscript clarifies the novelty and benefits of SUNLE.
>
>
> **About the inclusion of SMNLE in the figures**, we thank the reviewer for raising that point. Our updated manuscript now shows in Figure 2 the results of SMNLE when applicable (e.g. for the amortized inference setting only).

---

> > ### Comment · Reviewer_jDWH · 2022-12-05
> > **Reaction to Author Response**
> >
> > Thank you for revising your paper.
> > The fact that your fitting a conditional EBM instead of a joint is not a novel contribution. Importantly SUNLE still remains dependent on  Doubly-Intractable MCMC, and getting rid of Doubly-Intractable MCMC was a part of the motivation for the AUNLE!
> >
> > After reading the other reviews and responses, I am going to keep my previous rating.

---

> > > ### Author Response · Authors · 2022-12-09
> > > **Response to reviewer's  comments**
> > >
> > > We thank the reviewer for their additional comments. We address the reviewer's remarks below:
> > >
> > > **About conditional training in [1]:** We have clarified in our latest [answer](https://openreview.net/forum?id=gL68u5UuWa&noteId=3B8Sqc_Ug4) to reviewer kgMn the differences between our work and prior art on conditional EBMs. In short, while we found out, after additional investigation, that [1] indeed adopts a similar conditional training approach as ours. However, [1] described neither an objective function for training conditional EBMs, nor the way to approximate this objective's gradient, while we elucidate both these points in our paper. Also, while [1] considered only conditioning over discrete-valued variables, we show that it is possible to train conditional EBM with continuously-valued conditioned variables. We will make both points clearer in the final version of the manuscript.
> > >
> > >
> > > [1] Implicit Generation and Modeling with Energy-Based Models https://arxiv.org/pdf/1903.08689.pdf

---

### Official Review · Reviewer_mHnK · 2022-10-27

**Confidence:** 2
**Correctness:** 3
**Technical Novelty And Significance:** 3
**Empirical Novelty And Significance:** 3
**Recommendation:** 5

**Clarity, Quality, Novelty And Reproducibility:**

The paper is in general well written and easy to follow but since in the paper there are many algorithms with abbreviation. I do recommend the authors to try to make these more clear.

**Strength And Weaknesses:**

I mainly work on EBM instead of SBI. So my opinion my focus more on EBM part.

For the shining point of this paper, I think I like idea of this paper of using EBM to model the joint or conditional distribution of the paper. On one hand, EBM can be more expressive than normalizing flow given similar number of parameters and on the other hand, it makes sense to me that by cleverly design the EBM term, one can avoid the approximate doubly intractable estimation needed in previous method.

However, I also have some questions.
1. As pointed out by [2] and [4], EBM trained with MCMC as a sampler and MLE as objective may have the "short-run" effect. That is, instead of learning a correct energy function, the model converges to a point that the Langevin sampler becomes a noise injection generator. Since in the task of SBI, one actually needs a correct energy term to infer the likelihood. I'm wondering whether this problem exists in the current training process, especially for those complex tasks.

2. The authors mentioned that their model can be more effective comparing with previous methods. Thus, I think it is necessary to add a complexity analysis part for both AUNLE and SMNLE, e.g. list different models' (both the new models and baselines) accuracies together with their training/testing/inference time and make a comparison.

3. I'm not very sure whether the SBI experiments results listed here are strong enough to make conlusion that the new proposed method outperforms the previous ones. Would like to listen to other reviewer's opinions.

4. Some important EBM literatures are missing. Recommend the authors to add the following [1] [2] [4] papers.

[1]  A Theory of Generative ConvNet. ICML 2016

[2] Learning non-convergent nonpersistent short-run mcmc toward energy-based model.  NeurIPS, 2019

[3] Cooperative Training of Descriptor and Generator Networks. PAMI, 2020

[4] A Tale of Two Flows: Cooperative Learning of Langevin Flow and Normalizing Flow Toward Energy-Based Model. ICLR 2022

**Summary Of The Paper:**

This paper try to solve the Simulation-Based Inference (SBI) task using the energy-based model (EBM). In SBI task, one assume a known simulator that can generate samples x from interesting distribution parameterized by $\theta$. The task is then estimate $p(\theta | x)$. The authors using EBM to parameterize $q_\phi(x|\theta) $ or $q_\phi(x, \theta)$ and then inference can be done using bayes rule. In the paper, the authors propose two algorithms. AUNLE use a parameterization with tilting trick to make the normalizing constant of EBM independent of $\theta$ and thus reduce the computational cost in the previous SMNLE method. And AUNLE replaces the normalizing flow model used in previous method with EBM. Experimental results seem to support that the proposed models outperform its baselines.

**Summary Of The Review:**

In general, I like the idea of this paper of using EBM to solve the Simulation Based Inference (SBI) problem. And I think the theory part is OK for me.  But since I'm not an expert in SBI, I would like to listen other reviewers' opinion on how solid the experiments are.

---

> ### Author Response · Authors · 2022-11-18
> **Response to reviewer - thank you for your review**
>
> We thank the reviewer for their comments and suggestions. We address the reviewer's comments below:
>
> **Regarding the presence of a short run effect in UNLE**, we believe that this is an interesting point. Importantly, UNLE does not use short-run Langevin dynamics, but *long-run MCMC*, both for training and inference:
> 1. during training, we initialize the MCMC chain using a warm-starting strategy across iterations.
> 2. during inference, we run a MCMC chain during a burn-in period before collecting posterior samples.
>
> These two aspect contrast sharpely with short-run EBMs, that rely on short run Langevin dynamics for both training and inference. For that reason, *we believe that UNLE does not suffer from some short-run effect*. We mentioned the short-run effect in the main body of the paper, and discussed it in detail in Appendix B.2.
>
> *Additionally*, we added (in appendix B.2) a plot of UNLE's density for two-moons model, which possesses a 2-dimensional posterior than can be visualized easily. The figures showed that UNLE's posterior estimate approximates very faithfully the true density, and that UNLE is thus a faithful density estimator, unlike short-run EBMs.
>
> **Regarding the time complexity of training UNLE**: We agree that this is an important point. Our updated manuscript makes a detailed analysis of computational cost complexity of UNLE in Appendix B.4. We rely on an efficient JIT-compiled MCMC implementation to perform gradient approximation, and make use of several warm-starting strategies to decrease the computational cost of UNLE during both training and inference. **All in all, UNLE is the fastest method among all benchmarked methods**. 10 rounds of SUNLE for the pyloric network takes a total of 3 hours (with half this time spent on simulating data), and is thus *10 times faster than SNVI, the prior state-of-the-art on this problem*.
>
>
> **Regarding the ambition of our experiments**, we had two objectives:
> 1. Illustrate the behavior of our method in well-understood environments.
> 2. Apply our method to a challenging real-world example.
>
> For the first objective, we indeed followed a widely used experimental setup in the SBI literature:  *the SBI benchmark suite* (Lueckmann et. al.; 2021), a benchmark suite with challenging posteriors (see Appendix B.1). We emphasize that many milestone SBI publications like  NLE (Papamakarios et al., 2019), NPE (Greenberg et al., 2019) reported results on benchmark models as they are well understood, and allow for a fair comparison between methods.
>
> For the second objective, we used *a real-world example on a full-scale simulator used in computational neuroscience*. The real-world experiment tackles a high-impact problem, with numerous highly cited works (Prinz et al., 2003; 2004; Haddad & Marder) dedicated to solving inference and simulation for this model already published. Inference for this model is challenging: only a narrow region of the prior support yields valid neural dynamics, with state of the art methods achieving only 85% of such dynamics on final posterior samples. Our proposed method significantly improves the quality of inference upon prior methods, and we thus believe that our method could benefit domain scientists for inference tasks with complex likelihoods in a domain where normalizing flows constitute the dominant modeling choice.
>
> **About the missing references in the EBM literature**: we thank the reviewer for pointing us to such works:
>
> - [2] and [4] are related to the short-run effect of EBMs; in particular, [4] exploits the short-time effect of MCMC to interpret the resulting model as a flow-like model and proposes a cooperative strategy for learning it in combination with a normalizing flow. We refer to them in the paragraph of our submission discussing the potential manifestation of this effect in UNLE.
> -  Our understanding is that [3]  introduce a training method for an image generation model with two components: a EBM and a generator network, while [1] introduces an EBM model called generative ConvNet that is used for natural image generation. Thus, we could not find a clear way to tie these works to ours, as our goal is neither to review all EBM literature nor to focus on specific energy network architectures. We would appreciate recommendations from the reviewer as of how to tie [1] and [3] to our work.

---

> > ### Comment · Reviewer_mHnK · 2022-12-07
> > **I lean to keep my original rating**
> >
> > I would like to first thank the authors for their detailed rebuttal. This does solve some of my concerns, but not all. After reading other reviewers opinion, I current still have the following concerns:
> > 1. For the short-run effect, the new comment by reviewer kgMn points out the remaining possible problem, which makes sense to me;
> > 2. Other reviewers (kgMn, jDWH) raise (and still hold) concerns regarding the novelty of this paper;
> > 3. Also, it seems that besides me, other reviewers (kgMn and maybe VwrC) also have concerns regarding whether the experiments are too limited to demonstrate the effectiveness of the proposed model.
> > Thus I would keep my rating as botherline reject.
> >
> > PS: For the related work, I include [1] [3] here because I think they are all milestones of EBM. For example, [1] is perhaps the first paper that uses deep neural network as energy function and Langevin Dynamics as sampling methods. [3] is an extension of [1] and it is the foundation framework of [4]. Thus, I think should be included in section 2.1, the introduction of EBM, together with the other papers like [1].

---

> > > ### Author Response · Authors · 2022-12-09
> > > **Response to reviewer's comments**
> > >
> > > Thank you for your answer. We will add [1] and [3] in the introduction of EBMs.
> > >
> > > **About the novelty of SUNLE:** We have clarified in our latest [answer](https://openreview.net/forum?id=gL68u5UuWa&noteId=3B8Sqc_Ug4) to reviewer kgMn the differences between our work and prior art on conditional EBMs. In short, while we found out, after additional investigation, that [6] indeed adopts a similar conditional training approach as ours. However, [6] described neither an objective function for training conditional EBMs, nor a way to approximate the objective's gradient, while we elucidate both these points in our paper. Also, while [6] considered only conditioning over discrete-valued variables, we show that it is possible to train conditional EBM with continuously-valued conditioned variables. We will make both points clearer in the final version of the manuscript.
> > >
> > >
> > > **About the short-run effect** We agree with the reviewer that it is possible that the short run effect manifests itself in the case of very high-dimensinonal models; however, many applications of SBI involve require only dealing with variables of moderate dimension thanks to expert knowledge. But we agree that investigating short-time effect of MCMC in the context of SBI would be an interesting direction for future work. We would like to point out to the reviewers that our training approach follows the recommendations of [5] for long-run EBMs: In particular, we use a consistent MCMC sampler (akin to setting $\tau=1$ in [5] and adding a Metropolis-Hasting correction step), we adapt the MCMC proposal step size $\epsilon$ **in an adaptive manner during training**, and we preform persistent initialization. We will make these points clearer in the paper.
> > >
> > >
> > > **About the experiments** Simulation-Based Inference aims at performing inference in domain sciences, like neuroscience, ecology or particle physics. In such problems, the random variables of interest can be high dimensional, but are often reduced to lower-dimensional variables thanks to summary statistics crafted by domain experts: this is on the said lower-dimensional statistics that SBI models are trained on. The SBI setting thus differs from traditional Computer Vision or NLP settings.
> > >
> > > [5] On the anatomy of MCMC-based maximum likelihood learning with energy-based models. https://arxiv.org/pdf/1903.12370.pdf
> > >
> > > [6] Implicit Generation and Modeling with Energy-Based Models https://arxiv.org/pdf/1903.08689.pdf

---

### Official Review · Reviewer_VwrC · 2022-11-04

**Confidence:** 4
**Correctness:** 4
**Technical Novelty And Significance:** 3
**Empirical Novelty And Significance:** 2
**Recommendation:** 6

**Clarity, Quality, Novelty And Reproducibility:**

The paper is very well written, has clarity on the details of the techniques and makes sense.
It also nicely ties into prior work on SBI and positions the paper fairly.
The main contribution is also fairly interesting as it reduces the computational budget for SBI, which is a core goal in the field.

**Strength And Weaknesses:**

Stregths:
- the authors propose an interesting "tilted" inference procedure which performs well and is a neat trick
- the utilization of conditional EBMs in SBI makes sense and is well executed
- the authors propose a useful sequential version of their system

Weaknesses:
- like in many papers on this topic, the experiments are quite toy and not as ambitious as they could be, but they are clean and well-executed.

**Summary Of The Paper:**

The authors propose two new synthetic likelihood methods for simulation based inference using energy based models, UNLE and AUNLE.

They introduces a tilting trick and an amortized sequential model to perform posterior inference with intractable likelihoods and improve both modeling quality and inference budget.

**Summary Of The Review:**

The authors propose a rigorous and interetsting spin on SBI using EBMs.

The proposed extensions make sense and should be useful in the field, and the experiments appear well justified -if comparatively simple.

---

> ### Author Response · Authors · 2022-11-18
> **Response to reviewer - thank you for your review.**
>
> We thank the reviewer for their positive feedback and suggestions. We are glad you found the tilting method interesting and the utilization of conditional EBMs relevant and well-executed. We also thank you for noticing our effort to make clean and fair comparisons in the experiments.
>
> **Regarding the ambition of our experiments**, we had two objectives:
> 1. Illustrate the behavior of our method in well-understood environments
> 2. Apply our method to a challenging real-world example.
>
> For the first objective, we indeed followed a widely used experimental setup in the SBI literature:  *the standard SBI benchmark suite*. We emphasize that many milestone SBI publications like  NLE (Papamakarios et al., 2019) and NPE (Greenberg et al., 2019) reported results on benchmark models as they are well understood, and allow for a fair comparison between methods.
>
> For the second objective, we used *a real-world example on a full-scale simulator used in computational neuroscience*. The real-world experiment tackles a high-impact problem, with numerous highly cited works (Prinz et al., 2003; 2004; Haddad & Marder) dedicated to solving inference and simulation for this model already published. Inference for this model is challenging: only a narrow region (1%) of the prior support yields valid neural dynamics, with state of the art methods achieving only 85% of such dynamics on final posterior samples. Our proposed method significantly improves the training speed and quality of inference upon prior methods.

---

### Author Response · Authors · 2022-11-18
**Post-rebuttal summary by authors**

We thank the reviewers for their careful review and insightful comments and suggestions.
We answered each reviewer individually, and will gladly answer any remaining questions. We synthesize the main points brought by reviewers, and the changes we made to the manuscript to address them.

**Updates on experiments**

- All experiments in the manuscripts are now **averaged across 5 random seeds**. We observe little variance across seeds, highlighting the robustness of our methods.
- We made implementations improvements to SUNLE which have **significantly increased the performance and stability of the procedure, while reducing its total runtime**. SUNLE achieves now significantly better results than SNVI on the pyloric network model, **with a final steady rate of 92% of valid simulations stemming from posterior samples**, compared to 84% for SNVI. Additionally, SUNLE is trained in *less than 3 hours (with half of it spent simulating samples), which is **10 times faster** than SNVI*. We added a more comprehensive runtime analysis in the appendix.
- The updated plots now compare AUNLE with SMNLE (see Figure 2), the other unnormalized likelihood method for amortized SBI. AUNLE significantly outperforms SMNLE in all settings. We detail the experimental setup used in our experiments in appendix. Note than SMNLE does not have a sequential version, and is thus not tested for sequential inference on the pyloric model.
- We made additional checks (see Appendix B.3) to verify that the uniformization property of AUNLE's posterior holds in practice. We also add a discussion on the (absence of) short-run effect on UNLE's posterior.


**Updates on SUNLE**

Multiple reviewers have commented on the similarity between SUNLE and existing EBM training methods. Our updated manuscript now highlights the benefits and novelty of SUNLE over existing conditional EBM training methods:

- SUNLE's objective directly targets its conditional density model $q_\psi(x|\theta)$. Unlike known objectives for EBMs which target a joint model, this objective does not require modelling and learning the proposal $\pi$, which is analytically intractable in the sequential setting, and is not needed for inference. Bypassing the need for relearning the proposal is important when the proposal is highly complex, as is often the case in SBI when it is set to be the current posterior estimate. Our updated manuscript contains a clearer motivation and explanation of SUNLE's algorithm in Section 3.2.
- SUNLE's objective departs from known objectives on joint models, and requires a new training algorithm, which differs from training joint models using Contrastive Divergence in ways that we detail in our manuscript.


We have updated our manuscript accordingly to account for all points discussed above: all additions made to the paper during the rebuttal period are highlighted in red. We refer to the updated manuscript, and individual answers to reviewers for additional details. We hope that these clarifications will help the reviewers in their assessment of the paper, and we are happy to answer any remaining questions.

---

### Decision · Program_Chairs · 2023-01-20

**Decision:**

Reject

**Justification For Why Not Higher Score:**

the current paper has a major issue about marginal novelty, which directly leads to a rejection of the paper at the current stage.

**Justification For Why Not Lower Score:**

N/A

**Metareview: Summary, Strengths And Weaknesses:**

This paper studies a novel application of energy-based models for simulation-based inference, and experiments are conducted to verify the effectiveness of the proposed framework. The paper is clear, well-written and organized. The rebuttal only addressed some of the concerns raised by 5 reviewers and the paper finally received 5 mixed scores (i.e., 3 marginal reject, 1 marginal accept, and 1 accept). After the rebuttal, the average score falls into the range of a borderline paper.  Even though Reviewer m5HY champions the paper due to the novelty, there still remains unaddressed concerns raised by other reviewers. For example, Reviewers mHnK, kgMn and jDWH think that the novelty of the paper is marginal because the training of the EBM is standard and only the application is new. Besides, Reviewers mHnK, kgMn and VwrC think that the experiments are too limited to demonstrate the effectiveness of the proposed model. After reading the rebuttal and having an internal discussion with the reviewers, the AC thinks that even though some of the reviewers are satisfied with the rebuttal or increase their rating due to the feedback from the authors, the current paper still has critical concerns about limited novelty and insufficient empirical results. Therefore, the AC has to recommend rejecting the paper at the current stage. However, due to the promising outputs and the interesting applications of EBM shown in the paper, the AC urges the authors to improve and revise their work by considering the reviewers’ comments and re-submit it to the next venue in the future.


**Summary Of Ac-Reviewer Meeting:**

Because some of the reviewers are unresponsive, it was hard to gather all of them in a virtual meeting, so that the AC has to use the forum to discuss with the reviewers and meet some reviewers who are responsive and willing to meet in a virtual meeting.

Besides those final comments, judgements and discussion that are visible in the review system, Reviewer mHnKr shares his concern on the limited novelty for the current paper based on his expertise and experience on energy-based modeling and learning.

Although Reviewer m5HY champions the paper due to the novelty, with a score of 8, but from his/her review comments, the AC could not find any strong or useful information about his/her justification on the paper novelty. Thus, the AC downweights the rating of Reviewer mHnKr, and takes into serious account the concerns on novelty from Reviewers mHnK, kgMn and jDWH. The AC recommends rejecting the paper.